# NON-PARAMETRIC NEURO-ADAPTIVE CONTROL

## ABSTRACT

We develop a learning-based algorithm for the control of autonomous systems governed by unknown, nonlinear dynamics to satisfy user-specified tasks expressed via time-varying reference trajectories. Most existing algorithms either assume certain parametric forms for the unknown dynamic terms or resort to unnecessarily large control inputs in order to provide theoretical guarantees. The proposed algorithm addresses these drawbacks by integrating neural-network-based learning with adaptive control. More specifically, the algorithm learns a controller, represented as a neural network, using training data that correspond to a collection of system parameters and tasks. These parameters and tasks are derived by varying the nominal parameters and the reference trajectories, respectively. It then incorporates this neural network into an online closed-form adaptive control policy in such a way that the resulting behavior satisfies the user-defined task. The proposed algorithm does not use any a priori information on the unknown dynamic terms or any approximation schemes. We provide formal theoretical guarantees on the satisfaction of the task. Numerical experiments on a robotic manipulator and a unicycle robot demonstrate that the proposed algorithm guarantees the satisfaction of 50 user-defined tasks, and outperforms control policies that do not employ online adaptation or the neural-network controller. Finally, we show that the proposed algorithm achieves greater performance than standard reinforcement-learning algorithms in the pendulum benchmarking environment.

## 1 INTRODUCTION

Learning and control of autonomous systems with uncertain dynamics is a critical and challenging topic that has been widely studied during the last decades. One can identify plenty of motivating reasons, ranging from uncertain geometrical or dynamical parameters and unknown exogenous disturbances to abrupt faults that significantly modify the dynamics. There has been, therefore, an increasing need for developing control algorithms that do not rely on the underlying system dynamics. At the same time, such algorithms can be easily implemented on different, heterogeneous systems, since one does not need to be occupied with the tedious computation of the dynamic terms.

A promising step towards the control of systems with uncertain dynamics is the use of data obtained a priori from system runs. However, engineering systems often undergo purposeful modifications (e.g., substitution of a motor or link in a robotic arm or exposure to new working environments) or suffer gradual faults (e.g., mechanical degradation), which might change the systems' dynamics or operating conditions. Therefore, one cannot rely on the aforementioned data to provably guarantee the successful control of the system. On the other hand, the exact incorporation of these changes in the dynamic model, and consequently, the design of new model-based algorithms, can be a challenging and often impossible procedure. Hence, the goal in such cases is to exploit the data obtained a priori and construct intelligent online policies that achieve a user-defined task while adapting to the aforementioned changes.

There has been a large variety of works that tackle the problem of control of autonomous systems with uncertain dynamics, exhibiting, however, certain limitations. The existing algorithms are based on adaptive and learning-based approaches or the so-called funnel control (Krstic et al. (1995); Kanellakopoulos et al. (1991); Xu & Ioannou (2003); Ge & Wang (2004); Wen et al. (2011); Chen et al. (2011); Slotine & Li (1987); Slotine & Coetsee (1986); Chen et al. (2008); Xian et al. (2004); Vamvoudakis & Lewis (2010); Berger et al. (2018); Bechlioulis & Rovithakis (2014); Joshi et al. (2020); Capotondi et al. (2020); Bertsekas & Tsitsiklis (1996); Sutton & Barto (2018)). Nevertheless,

adaptive control methodologies are restricted to system dynamics that can be linearly parameterized with respect to certain unknown parameters (e.g., masses, moments of inertia), assuming the system structure perfectly known; funnel controllers employ reciprocal terms that drive the control input to infinity when the tracking error approaches a pre-specified funnel function, creating thus unnecessarily large control inputs that might damage the system actuators. Data-based learning approaches either consider some system characteristic known (e.g., a nominal model, Lipschitz constants, or global bounds), or use neural networks to learn a tracking controller or the system dynamics; the correctness of such methods, however, relies on strong assumptions on the parametric approximation by the neural network and knowledge of the underlying radial basis functions. Finally, standard reinforcement-learning techniques (Bertsekas & Tsitsiklis (1996); Sutton & Barto (2018)) usually assume certain state and/or time discretizations of the system and rely on exhaustive search of the state space, which might lead to undesirable transient properties (e.g., collision with obstacles while learning).

## 1.1 CONTRIBUTIONS AND SIGNIFICANCE

This paper addresses the control of systems with continuous, *unknown* nonlinear dynamics subject to tasks expressed via time-varying reference trajectories. Our main contribution lies in the development of a learning-based control algorithm that guarantees the accomplishment of a given task using only mild assumptions on the system dynamics. The algorithm draws a novel connection between adaptive control and learning with neural network representations, and consists of the following steps. Firstly, it trains a neural network that aims to learn a controller that accomplishes a given task from data obtained off-line. Secondly, we develop an online adaptive feedback control policy that uses the trained network to guarantee convergence to the given reference trajectory and hence satisfaction of the task. Essentially, our approach builds on a combination of off-line trained controllers and on-line adaptations, which was recently shown to significantly enhance performance with respect to single use of the off-line part (Bertsekas (2021)).

The major significance of our contribution is twofold. Firstly, we guarantee the theoretical correctness of the proposed algorithm by considering only mild conditions on the neural network, removing the long-standing assumptions on parametric approximations and boundedness of the estimation error. Secondly, we demonstrate via the experimental results the generality of the algorithm with respect to different tasks and system parameters. That is, the training data that we generate for the training of the neural network in the first step correspond to tasks that are different from the given one to be executed[1]. Additionally, we employ systems with different dynamic parameters to generate these data. We evaluate the proposed algorithm in numerous scenarios comprising different variations of the given task and different system dynamic parameters, which do not necessarily match the training data. We show that the algorithm, owing to its adaptation properties, is able to guarantee the satisfaction of the respective tasks in all the aforementioned scenarios by using the same neural network.

## 1.2 RELATED WORK

A large variety of previous works considers neuro-adaptive control with stability guarantees, focusing on the optimal control problem (Vamvoudakis & Lewis (2010); Yang et al. (2020); Cheng et al. (2007); Fan et al. (2018); Kiumarsi et al. (2017); Zhao & Gan (2020); Vrabie & Lewis (2009); Sun & Vamvoudakis (2020); Kamalapurkar et al. (2015); Huang et al. (2018b); Mo et al. (2019); Joshi et al. (2020)). Nevertheless, the related works draw motivation from the neural network density property (see, e.g., (Cybenko (1989)))[2] and assume sufficiently small approximation errors and linear parameterizations of the unknown terms of the form $W(x)\theta$ (dynamics, optimal controllers, or value functions), with $W$ and $\theta$ known and unknown, respectively. Similarly, more traditional adaptive control methodologies that handle uncertain nonlinear systems assume either linear parameterizations of the unknown dynamic terms (Krstic et al. (1995); Hong et al. (2009); Chen (2019); Huang et al. (2018a); Kanellakopoulos et al. (1991); Xu & Ioannou (2003); Ge & Wang (2004); Wen et al. (2011); Chen et al. (2011); Slotine & Li (1987); Slotine & Coetsee (1986); Chen et al. (2008)), use known upper-bound functions (Slotine & Coetsee (1986)), or provide local stability results dictated by the dynamic bounds (Xian et al. (2004); Chen et al. (2008)). This paper relaxes the aforementioned assumptions and proposes a *non-parametric* neuro-adaptive controller, whose stability guarantees

---

[1] The task difference is illustrated in Section 4.

[2] A sufficiently large neural network can approximate a continuous function arbitrarily well in a compact set.

rely on a mild boundedness condition of the closed-loop system state that is driven by the learned controller. The proposed approach exhibits similarities with (Liu et al. (1994)), which employs off-line-trained neural networks with online feedback control, but fails to provide convergence guarantees. Similarly, Zeng et al. (2020) develops an algorithm that combines a learning module and a physics-based controller for robot throwing of arbitrary objects, assuming however access to a physics simulator and not providing theoretical guarantees.

Other learning-based related works include modeling with Gaussian processes (Capotondi et al. (2020); Leahy et al. (2019); Jain et al. (2018); Berkenkamp & Schoellig (2015)), or use neural networks (Ma et al. (2020); Shah et al. (2018); Yan & Julius (2021); Liu et al. (2021); Hahn et al. (2020); Cai et al. (2021); Wang et al. (2020); Camacho & McIlraith (2019); Hahn et al. (2020); Riegel et al. (2020); Hu et al. (2020); Ivanov et al. (2019)) to accomplish reachability, verification or temporal logic specifications. Nevertheless, the aforementioned works either use partial information on the underlying system dynamics, or do not consider them at all. In addition, works based on Gaussian processes usually propagate the dynamic uncertainties, possibly leading to conservative results. Similarly, data-driven model-predictive control techniques (Nubert et al. (2020); Maddalena et al. (2020)) use data to over-approximate additive disturbances or are restricted to linear systems.

Control of unknown nonlinear continuous-time systems has been also tackled in the literature by using funnel control, without necessarily using off-line data or dynamic approximations (Berger et al. (2018); Bechlioulis & Rovithakis (2014); Lindemann et al. (2017); Verginis & Dimarogonas (2018); Verginis et al. (2021)). Nevertheless, funnel controllers usually depend on reciprocal time-varying barrier functions that drive the control input to infinity when the error approaches a pre-specified funnel, creating thus unnecessarily large control inputs that might damage the system actuators.

## 2 PROBLEM FORMULATION

Consider a continuous-time dynamical system governed by the 2nd-order continuous-time dynamics

$$\ddot{x} = f(\bar{x}, t) + g(\bar{x}, t)u(\bar{x}, t), \tag{1}$$

where $\bar{x} := [x^\top, \dot{x}^\top]^\top \in \mathbb{R}^{2n}$, $n \in \mathbb{N}$, is the system state, assumed available for measurement, and $u : \mathbb{R}^{2n} \times [0, \infty) \to \mathbb{R}^n$ is the time-varying feedback-control input. The terms $f(\cdot)$ and $g(\cdot)$ are nonlinear vector fields that are locally Lipschitz in $\bar{x}$ over $\mathbb{R}^{2n}$ for each fixed $t \geq 0$, and uniformly bounded in $t$ over $[0, \infty)$ for each fixed $\bar{x} \in \mathbb{R}^{2n}$. The dynamics (1) comprise a large class of nonlinear dynamical systems (Zhong & Leonard (2020); Yu et al. (1996); Doya (1997)) that capture contemporary engineering problems in mechanical, electromechanical and power electronics applications, such as rigid/flexible robots, induction motors and DC-to-DC converters, to name a few. The continuity in time and state provides a direct link to the actual underlying system, and we further do not require any time or state discretizations.

We consider that $f(\cdot)$ and $g(\cdot)$ are completely unknown; we do not assume any knowledge of the structure, Lipschitz constants, or bounds, and we do not use any scheme to approximate them. Note also that we do not assume *global* Lipschitz continuity or global boundedness of $f(\cdot, t)$ and $g(\cdot, t)$ or the solution $\bar{x}(t)$ of (1). Nevertheless, we do assume that $g(\bar{x}, t)$ is positive definite:

**Assumption 1.** *The matrix $g(\bar{x}, t)$ is positive definite, for all $(\bar{x}, t) \in \mathbb{R}^{2n} \times [0, \infty)$.*

Such assumption is a sufficiently controllability condition for (1); intuitively, it states that the multiplier of $u$ (the input matrix) does not change the direction imposed to the system by the underlying control algorithm. Systems not covered by (1) or Assumption 1 consist of underactuated or non-holonomic systems, such as unicycle robots or underactuated aerial vehicles. Nevertheless, we provide an extension of our results for a non-holonomic unicycle vehicle in Appendix B. Moreover, the 2nd-order model (1) can be easily extended to account for higher-order integrator systems (Slotine et al. (1991)).

Consider now a time-constrained task expressed as a time-varying reference trajectory $p_\mathrm{d} : \mathbb{R}_{\geq 0} \to \mathbb{R}^n$. The objective of this paper is to construct a time-varying feedback-control algorithm $u(\bar{x}, t)$ such that the state of the closed-loop system (1) asymptotically tracks $p_\mathrm{d}$, i.e., $\lim_{t \to \infty}(x(t) - p_\mathrm{d}(t)) = 0$.

## 3 MAIN RESULTS

This section describes the proposed algorithm, which consists of two steps. Firstly, it learns a controller, represented as a neural network, using training data that correspond to a collection of different tasks and system parameters. Secondly, we design an adaptive, time-varying feedback controller that uses the neural-network approximation and guarantees tracking of the reference trajectory, consequently achieving satisfaction of the task.

### 3.1 NEURAL-NETWORK LEARNING

As discussed in Section 1, we are inspired by cases where systems undergo changes that modify their dynamics and hence the underlying controllers no longer guarantee the satisfaction of a specific task. In such cases, instead of carrying out the challenging and tedious procedure of identification of the new dynamic models and design of new model-based controllers, we aim to exploit data from off-line system trajectories and develop an online policy that is able to adapt to the aforementioned changes and achieve the task expressed via $p_{\mathrm{d}}$. Consequently, we assume the existence of offline data from a finite set of $T$ system trajectories that satisfy a collection of tasks, corresponding to bounded reference trajectories, including $p_{\mathrm{d}}$, and possibly produced by systems with different dynamic parameters. The data from each trajectory $i \in \{1, \dots, T\}$ comprise a finite set of triplets $\{\bar{x}_s(t), t, u_s(t)\}_{t \in \mathcal{T}_i}$, where $\mathcal{T}_i$ is a finite set of time instants, $\bar{x}_s(t) \in \mathbb{R}^{2n}$ are system states, and $u_s(t) \in \mathbb{R}^n$ are the respective control inputs, compliant with the dynamics (1). We use the data to train a neural network in order to approximate the respective controller $u(\bar{x}, t)$. More specifically, we use the pairs $(\bar{x}_s(t), t)_{t \in \mathcal{T}_i}$ as input to a neural network, and $u_s(t)_{t \in \mathcal{T}_i}$ as the respective output targets, for all trajectories $i \in \{1, \dots, T\}$. For given $\bar{x} \in \mathbb{R}^{2n}, t \in \mathbb{R}_{\geq 0}$, we denote by $u_{\mathrm{nn}}(\bar{x}, t)$ the output of the neural network. Note that the controller $u(\bar{x}, t)$, which the neural network aims to approximate, is not associated to the specific task expressed via $p_{\mathrm{d}}$ and mentioned in Section 2, but a collection of several tasks. Therefore, we do not expect the neural network to learn how to track $p_{\mathrm{d}}$, but rather to be able to adapt to the entire collection of tasks. This is an important attribute of the proposed scheme, since it can generalize over tasks. The motivation for training the neural network with different tasks and dynamic parameters is the following. Since the tasks correspond to bounded trajectories, the respective stabilizing controllers compensate successfully the dynamics in (1). Therefore, as will be clarified in the next section, the neural network aims to approximate an "average" controller the retains this property, i.e., the boundedness of the dynamics of (1). By using such approximation, the online feedback-control policy - illustrated in the next section - is able to guarantee tracking of $p_{\mathrm{d}}$ without using any explicit information on the dynamics.

### 3.2 FEEDBACK CONTROL DESIGN

As mentioned in Section 3.1, we do not expect the neural-network controller to accomplish tracking of $p_{\mathrm{d}}$, since the system (1) is trained on potentially different tasks and different system parameters, and (2) the neural network provides only an *approximation* of a stabilizing controller; potential deviations in certain regions of the state space might lead to instability. Moreover, the neural-network controller has no error feedback with respect to the open-loop trajectory $p_{\mathrm{d}}$; such feedback is substantial in the stability of control systems with dynamic uncertainties. Therefore, this section is devoted to the design of a feedback-control policy to track the trajectory $p_{\mathrm{d}}(t)$ by using the output of the trained neural network (see Fig. 1). The goal is to drive the error $e := x - p_{\mathrm{d}}$ to zero. As mentioned in Section 3.1, the motivation for training the neural network with several different tasks and dynamic parameters is the learning of a controller that is able to retain the property of the training controllers to compensate the system dynamics (1). This is officially stated in the following assumption regarding the closed-loop system trajectory that is driven by the neural network's output.

**Assumption 2.** *The output $u_{\mathrm{nn}}(\bar{x}, t)$ of the trained neural network satisfies*

$$\|f(\bar{x}, t) + g(\bar{x}, t)u_{\mathrm{nn}}(\bar{x}, t)\| \leq d\|\bar{x}\| + B \qquad (2)$$

*for positive constants $d$, $B$, for all $\bar{x} \in \mathbb{R}^{2n}$, $t \geq 0$.*

Intuitively, Assumption 2 states that the neural-network controller $u_{\mathrm{nn}}(\bar{x}, t)$ is able to maintain the *boundedness* of the system state by the constants $d$, $B$, which are considered to be *unknown*. The assumption is motivated by the property of neural networks to approximate a continuous function

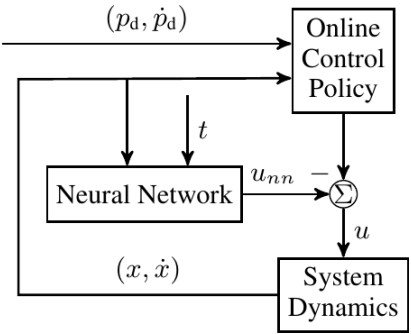

Figure 1: Block diagram of the proposed algorithm.

arbitrarily well in a compact domain for a large enough number of neurons and layers (Cybenko (1989))[3]. As mentioned before, since the collection of tasks, which the neural network is trained with, correspond to bounded trajectories, the system states are expected to remain bounded. Since $f(\bar{x}, t)$, and $g(\bar{x}, t)$ are continuous in $\bar{x}$ and bounded in $t$, they are also expected to be bounded as per (2). Contrary to the related works (e.g., (Vamvoudakis & Lewis (2010); Yang et al. (2020); Cheng et al. (2007); Fan et al. (2018))), however, we do not adopt approximation schemes for the system dynamics and we do not impose restrictions on the size of $d$, $B$. Moreover, Assumption 2 does not imply that the neural network controller $u_{\mathrm{nn}}(\bar{x}, t)$ guarantees tracking of the open-loop trajectory $p_{\mathrm{d}}$. Instead, it is merely a growth condition. Additionally, note that inequality 2 does not depend specifically on any of the tasks that the neural network is trained with. We exploit this property in the control design and achieve task generalization; that is, the open-loop trajectory $p_{\mathrm{d}}$ to be tracked (corresponding to the task $\varphi$) can be any of the tasks that the neural network is trained with.

We now define the feedback-control policy. Consider the adaptation variables $\hat{\ell}_1$, $\hat{\ell}_2$, corresponding to upper bounds of $d$, $B$ in (2), with $\hat{\ell}_1(0) > 0$, $\hat{\ell}_2(0) > 0$. We design first a reference signal for $\dot{x}$ as

$$v_{\mathrm{d}} := \dot{p}_{\mathrm{d}} - k_1 e, \tag{3}$$

that would stabilize the subsystem $\|e\|^2$, where $k_1$ is a positive control gain constant. Following the back-stepping methodology (Krstic et al. (1995)), we define next the respective error $e_v := \dot{x} - v_{\mathrm{d}}$ and design the neural-network-based adaptive control law as

$$u(\bar{x}, \hat{\ell}_1, \hat{\ell}_2, t) = u_{\mathrm{nn}}(\bar{x}, t) - k_2 e_v - \hat{\ell}_1 e_v - \hat{\ell}_2 \hat{e}_v \tag{4a}$$

$$\dot{\hat{\ell}}_1 = k_{\ell_1} \|e_v\|^2, \quad \dot{\hat{\ell}}_2 = k_{\ell_2} \|e_v\| \tag{4b}$$

where $k_2, k_{\ell_1}, k_{\ell_2}$ are positive constants, and $\hat{e}_v = \frac{e_v}{\|e_v\|}$ if $e_v \neq 0$, and $\hat{e}_v = 0$ if $e_v = 0$. The control design is inspired by adaptive control methodologies (Krstic et al. (1995)), where the time-varying gains $\hat{\ell}_1(t)$, $\hat{\ell}_2(t)$, adapt to the unknown dynamics and counteract the effect of $d$ and $B$ in (2) in order to ensure closed-loop stability. Note that the policy (3), (4) does not use any information on the system dynamics $f(\cdot)$, $g(\cdot)$ or the constants $B$, $d$. The tracking of $p_{\mathrm{d}}$ is guaranteed by the following theorem.

**Theorem 1.** *Let a system evolve according to (1) and let an open-loop trajectory $p_{\mathrm{d}} : \mathbb{R}_{\geq 0} \to \mathbb{R}^6$ encoding a user-defined task. Under Assumption 2, the control algorithm (4) guarantees $\lim_{t \to \infty}(e(t), e_v(t)) = 0$, as well as the boundedness of all closed-loop signals.*

We provide a sketch of the proof here and give the details in Appendix A.

*Proof (Sketch).* In view of (2), one can find positive constants $d_1$, $d_2$, $D$ such that

$$\frac{1}{\underline{g}}\|\underline{g}e + f(\bar{x}, t) + g(\bar{x}, t)u_{\mathrm{nn}}(\bar{x}, t) - \dot{v}_{\mathrm{d}}\| \leq d_1\|e\| + d_2\|e_v\| + D,$$

---

[3]For simplicity, we consider that (2) holds globally, but it can be extended to hold in a compact set.

for all $\bar{x} \in \mathbb{R}^{2n}$, $t \geq 0$, where $\underline{g}$ is the minimum eigenvalue of $g(\bar{x}, t)$, which is positive owing to the positive definitiveness of $g(\cdot)$. The constants $d_1$, $d_2$, and $D$ define upper bounds $\ell_1$ and $\ell_2$, which we aim to approximate by the adaptation variables $\hat{\ell}_1$ and $\hat{\ell}_2$, respectively. We next define the continuously differentiable function

$$V(\widetilde{x}) := \frac{1}{2}\|e\|^2 + \frac{1}{2\underline{g}}\|e_v\|^2 + \sum_{i \in \{1,2\}} \frac{1}{2k_{\ell_i}}(\hat{\ell}_i - \ell_i)^2$$

which represents an energy-like function of the system, and which we wish to drive to zero. One can prove, by analyzing the derivative $\dot{V}$ and using the control policy (4), that $V$ is non-decreasing along the solutions of the closed-loop system and that $\lim_{t\to\infty} \dot{V}(t) = 0$, which implies that $\lim_{t\to\infty} e(t) = \lim_{t\to\infty} e_v(t) = 0$. $\square$

Note that, contrary to works in the related literature (e.g., (Verginis & Dimarogonas (2020); Bechlioulis & Rovithakis (2014))), we do not impose reciprocal terms in the control input that grow unbounded in order to guarantee closed-loop stability. The resulting controller is essentially a simple linear feedback on $(e(t), e_v(t))$ with time-varying adaptive control gains, accompanied by the neural network output that ensures the growth condition (2). The positive gains $k_1$, $k_2$, $k_{\ell_1}$, $k_{\ell_2}$ do not affect the stability results of Theorem 1, but might affect the evolution of the closed-loop system; e.g., larger gains lead to faster convergence but possibly larger control inputs.

The proposed control algorithm does not require any of the long-standing assumptions on the system dynamics (1), such as linear parameterization, growth conditions, or boundedness by known functions (e.g., Krstic et al. (1995); Hong et al. (2009); Chen (2019); Huang et al. (2018a); Kanellakopoulos et al. (1991); Xu & Ioannou (2003); Ge & Wang (2004); Wen et al. (2011); Chen et al. (2011); Slotine & Li (1987); Slotine & Coetsee (1986); Chen et al. (2008); Slotine & Coetsee (1986); Chen et al. (2008); Xian et al. (2004)). Additionally, we do not assume the boundedness of the solution of (1) or of the dynamic terms $f(\cdot, t)$, $g(\cdot, t)$; instead, the control algorithm guarantees via Theorem 1 the boundedness of the system state as well as the asymptotic tracking of $p_d(t)$. The only boundedness condition that we require is (2) in Assumption 2, which can be accomplished by neural-network component in view of the universal approximation property Cybenko (1989).

## 4 NUMERICAL EXPERIMENTS

This section is devoted to a series of numerical experiments. More details can be found in Appendix B. We first test the proposed algorithm on a 6-dof UR5 robotic manipulator with dynamics

$$\ddot{x} = B(x)^{-1}\left(u - C(\bar{x})\dot{x} - g(x) + d(\bar{x}, t)\right) \tag{5}$$

where $x, \dot{x} \in \mathbb{R}^6$ are the vectors of robot joint angles and angular velocities, respectively; $B(x) \in \mathbb{R}^{6\times6}$ is the positive definite inertia matrix, $C(\bar{x}) \in \mathbb{R}^{6\times6}$ is the Coriolis matrix, $g(x) \in \mathbb{R}^6$ is the gravity vector, and $d(\bar{x}, t) \in \mathbb{R}^6$ is a vector of friction terms and exogenous time-varying disturbances.

The workspace consists of four points of interest $T_1$, $T_2$, $T_3$, $T_4$ (end-effector position and Euler-angle orientation), as depicted in Fig. 2, which correspond to the joint-angle vectors $c_1$, $c_2$, $c_3$, $c_4$. More information is provided in Appendix A. We consider a nominal task expressed via the spatio-temporal constraint $\bigwedge_{i \in \{1,...,4\}} G_{[0,\infty)} F_{I_i}(\|x_1 - c_i\| \leq 0.1)$, where $G$ and $F$ are the always and eventually operators respectively. The task consists of visits of $x_1$ to $c_i \in \mathbb{R}^6$ (within the radius 0.1) infinitely often within the time intervals dictated by $I_i$, for $i \in \{1, \ldots, 4\}$.

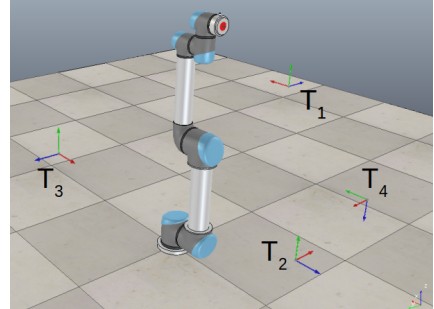

Figure 2: A UR5 robot in a workspace with four points of interest $T_i$, $i \in \{1, \ldots, 4\}$.

We create 150 problem instances by varying the positions of $c_i$, the time intervals $I_i$, the dynamic parameters of the robot (masses and moments of inertia of the robot's links and actuators), the friction and disturbance term $d(\cdot)$, the initial position and velocity of the robot, and the sequence of visits to the points $c_i$, as dictated by $\phi$, i.e., one instance

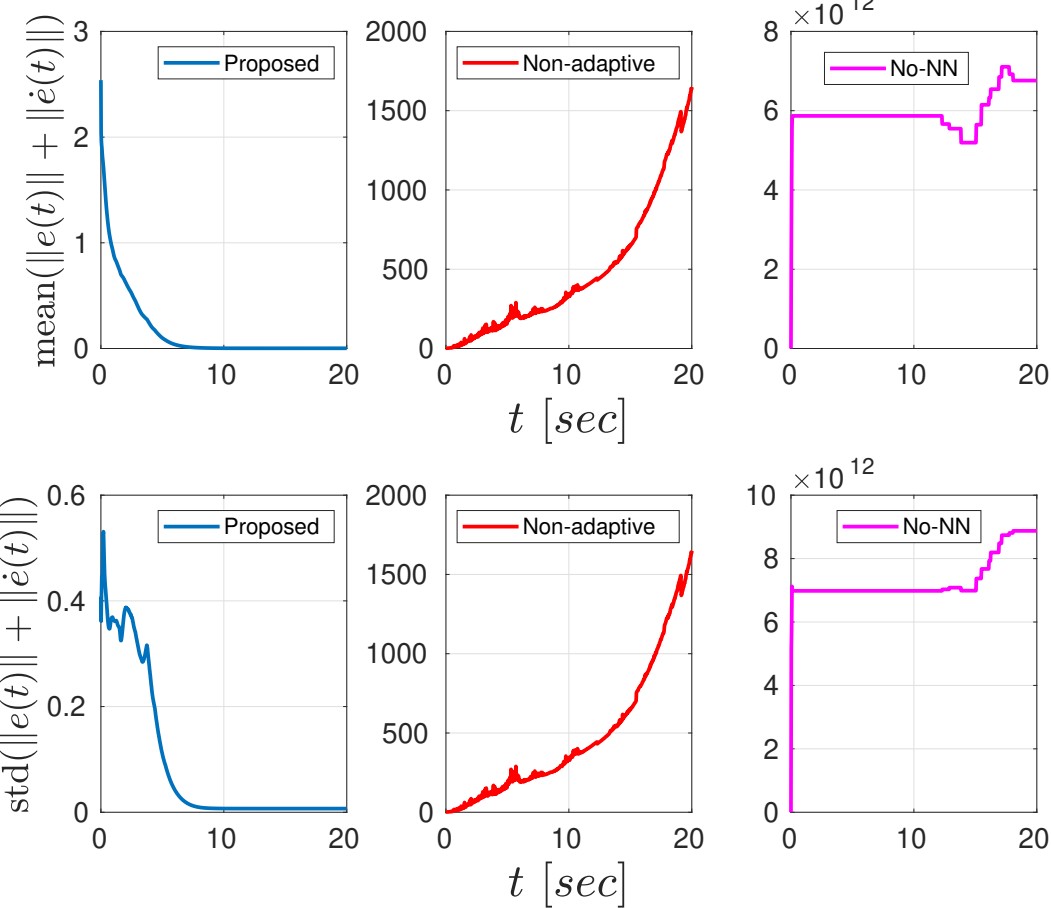

Figure 3: Mean (top) and standard deviation (bottom) of $\|e(t)\| + \|\dot{e}(t)\|$ for the proposed (left), non-adaptive (center), and no-neural-network (right) control policies.

might correspond to the visit sequence $((x(0), 0) \rightarrow (c_1, t_{1_1}) \rightarrow (c_2, t_{1_2}) \rightarrow (c_3, t_{1_3}) \rightarrow (c_4, t_{1_4})$, and another to $((x(0), 0) \rightarrow (c_3, t_{1_3}) \rightarrow (c_1, t_{1_1}) \rightarrow (c_4, t_{1_4}) \rightarrow (c_2, t_{1_2}))$. We separate the aforementioned 150 problem instances into 100 training instances and 50 test instances. We generate trajectories using the 100 training instances from system runs that satisfy different variations of one cycle of the task (i.e., one visit to each point). Each trajectory consists of 500 points and is generated using a nominal model-based controller. We use these trajectories to train a neural network and we test the control policy (4) in the 50 test instances. We also compare our algorithm with the non-adaptive controller $u_c(\bar{x}, t) = u_{\mathrm{nn}}(x, t) - k_1 e - k_2 \dot{e}$, as well as with a modified version $u_d(\bar{x}, t)$ of (4) that does not employ the neural network (i.e., the term $u_{\mathrm{nn}}(\bar{x}, t)$). The comparison results are depicted in Fig. 3, which depicts the mean and standard deviation of the signal $\|e(t)\| + \|\dot{e}(t)\|$ for the 50 instances and 20 seconds. It is clear from the figure that the proposed algorithm performs better than the non-adaptive and no-neural-network policies both in terms of convergence speed and steady-state error. It is worth noting that the non-adaptive policy results on average in unstable closed-loop system.

We next test the proposed algorithm, following a similar procedure, on a unicycle robot (the respective dynamics and control algorithm can be found in Appendix B). Fig. 5 depicts the mean and standard deviation of the errors $e_d(t)$, $e_\beta(t)$ for 50 test instances. We note that the performance of the no-neural-network control policy is much more similar to the proposed one than in the UR5 case. This can be attributed to (1) the lack of gravitational terms in the unicycle dynamics, which often lead to instability, and (2) the chosen control gains of the no-neural-network policy, which are sufficiently large to counteract the effect of the dynamic uncertainties.

Finally, we compare the performance of the proposed control policy with the reported data of (Wang et al. (2019)) on the benchmarking enivronment of the *pendulum*, where a single-link mechanical structure aims to reach the upright position. Following (Wang et al. (2019)), the reward and costs are $r_{pend}(t) = -\cos q(t) - 0.1\sin q(t) - 0.1\dot{q}(t) - 0.001u(t)^2$ and $J_{pend} = \sum_{t=1}^{H}\gamma^t r_{pend}(t)$, respectively.

Similar to the previous cases and in contrast to Wang et al. (2019), we generate 150 instances by varying the system parameters (pendulum length and mass), the external disturbances, and the initial conditions. We generate 100 trajectories, consisting of 500 points each, for the 100 training instances, by employing a nominal controller, and we use them train a neural network. Moreover, in order to guarantee the feasibility of the proposed algorithm (4) we consider larger control-action bounds than in (Wang et al. (2019)). The original bounds render these systems under-actuated, which is not included in the considered class of systems (1) and consist part of our future work. It should be noted that such larger bounds affect negatively the acquired rewards. We test the control policy (4) on the 50 test instances, for 5000 steps each, corresponding to 10 seconds. We set $H = 200$ and $\gamma = 1$ (Wang et al. (2019)). Fig. 4 depicts the

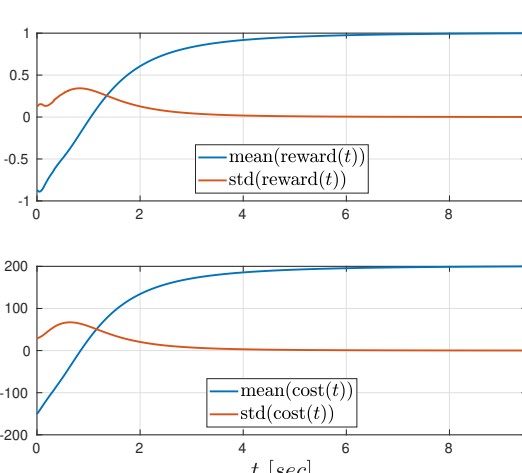

Figure 4: Mean and standard deviation of the reward and cost for the pendulum environment.

mean and standard deviation of the time-varying reward and cost functions, illustrating successful regulation to the upright position. After 5000 steps, we obtain a mean reward and cost of 0.99 and 200, respectively, showing better performance than the reported cost of 180 in (Wang et al. (2019)). Moreover, the proposed control algorithm achieves this performance without resorting to exhaustive exploration of the state-space, which is the case in the reinforcement-learning algorithms used in (Wang et al. (2019)). This is a very important property in practical engineering systems, where safety is of paramount significance and certain areas of the state space must be avoided.

## 5 DISCUSSION AND LIMITATIONS

As shown in the experimental results, the control algorithm is able to asymptotically track reference trajectories which were not considered when generating the training data. Similarly, the trajectories used in the training data were generated using systems with different dynamical parameters, and not specifically the ones used in the tests. The aforementioned attributes signify the ability of the proposed algorithm to generalize to different tasks and systems with different parameters. Nevertheless, the proposed control policy is currently limited by systems satisfying Assumption 1 and is not able to take into account under-actuated systems (e.g., the acrobot or cart-pole system); such systems consist part of our future work. Moreover, the proposed algorithm is dependent on the performance of the learning component to satisfy Assumption 2. Lack of satisfactory training might cause the neural-network output to add extra terms in the dynamics that render the closed-loop system unstable. A less conservative and more robust approach consists of using the off-line data to train multiple neural networks, each for a certain region of the state space; such an approach constitutes part of our future work. Finally, the discontinuities of (4), (12) might be problematic and create chattering when implemented in real actuators. A continuous approximation that has shown to yield satisfying performance is the boundary-layer technique (Slotine et al. (1991)).

## 6 CONCLUSION AND FUTURE WORK

We develop a novel control algorithm for the control of robotic systems with unknown nonlinear dynamics subject to tasks expressed as reference trajectories. The algorithm integrates neural network-based learning and adaptive control. We provide formal guarantees and perform extensive numerical

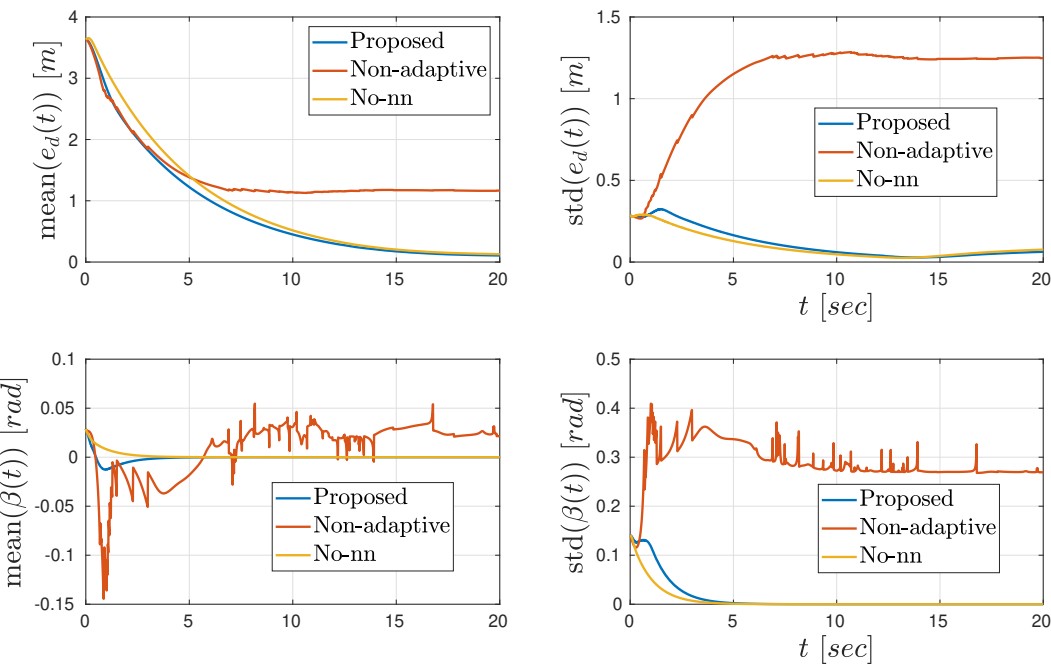

Figure 5: Mean (left) and standard deviation (right) of $e_d(t)$, $\beta(t)$ for the proposed, non-adaptive, and no-neural-network control policies.

experiments. Future directions will focus on relaxing the considered assumptions and extending the proposed methodology to underactuated systems.

## 7 REPRODUCIBILITY STATEMENT

We provide the proof of Theorem 1 in Appendix A. Additionally, we elaborate on the required assumptions (Assumptions 1, 2, 3) throughout the text - see pages 3, 5, and 7. Finally, we provide implementation details, instructions, and the respective code, required to reproduce the results, in Appendix C and the supplementary material.

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

## A   APPENDIX

We provide here the proof of Theorem 1. We first give some preliminary notation and background on systems with discontinuous dynamics.

NOTATION

Given a function $f : \mathbb{R}^n \to \mathbb{R}^k$, its Filippov regularization is defined as (Paden & Sastry (1987))

$$\mathsf{K}[f](x) := \bigcap_{\delta > 0} \bigcap_{\mu(\bar{N})=0} \overline{\mathrm{co}}(f(\mathcal{B}(x,\delta)\backslash\bar{N}), t), \tag{6}$$

where $\bigcap_{\mu(\bar{N})=0}$ is the intersection over all sets $\bar{N}$ of Lebesgue measure zero, $\overline{\mathrm{co}}(E)$ is the closure of the convex hull $\mathrm{co}(E)$ of the set $E$, and $\mathcal{B}(x,\delta)$ is the ball with radius $\delta$ centered at $x$.

NONSMOOTH ANALYSIS

Consider the following differential equation with a discontinuous right-hand side:

$$\dot{x} = f(x,t), \tag{7}$$

where $f : \mathcal{D} \times [t_0, \infty) \to \mathbb{R}^n$, $\mathcal{D} \subset \mathbb{R}^n$, is Lebesgue measurable and locally essentially bounded.

**Definition 1** (Def. 1 of (Fischer et al. (2013))). *A function $x : [t_0, t_1] \to \mathbb{R}^n$, with $t_1 > t_0$, is called a Filippov solution of (7) on $[t_0, t_1]$ if $x(t)$ is absolutely continuous and if, for almost all $t \in [t_0, t_1]$, it satisfies $\dot{x} \in \mathsf{K}[f](x, t)$, where $\mathsf{K}[f](x, t)$ is the Filippov regularization of $f(x, t)$.*

**Lemma 1** (Lemma 1 of (Fischer et al. (2013))). *Let $x(t)$ be a Filippov solution of (7) and $V : \mathcal{D} \times [t_0, t_1] \to \mathbb{R}$ be a locally Lipschitz, regular function[4]. Then $V(x(t), t)$ is absolutely continuous, $\dot{V}(x(t), t) = \frac{\partial}{\partial t} V(x(t), t)$ exists almost everywhere (a.e.), i.e., for almost all $t \in [t_0, t_1]$, and $\dot{V}(x(t), t) \overset{a.e}{\in} \dot{\tilde{V}}(x(t), t)$, where*

$$\dot{\tilde{V}} := \bigcap_{\xi \in \partial V(x,t)} \xi^\top \begin{bmatrix} \mathsf{K}[f](x,t) \\ 1 \end{bmatrix},$$

*and $\partial V(x, t)$ is Clarke's generalized gradient at $(x, t)$ (Fischer et al. (2013)).*

**Corollary 1** (Corollary 2 of (Fischer et al. (2013))). *For the system given in (7), let $\mathcal{D} \subset \mathbb{R}^n$ be an open and connected set containing $x = 0$ and suppose that $f$ is Lebesgue measurable and $x \mapsto f(x, t)$ is essentially locally bounded, uniformly in $t$. Let $V : \mathcal{D} \times [t_0, t_1] \to \mathbb{R}$ be locally Lipschitz and regular such that $W_1(x) \le V(x, t) \le W_2(x)$, $\forall t \in [t_0, t_1]$, $x \in \mathcal{D}$, and*

$$z \le -W(x(t)), \ \ \forall z \in \dot{\tilde{V}}(x(t), t), \ t \in [t_0, t_1], \ x \in \mathcal{D},$$

*where $W_1$ and $W_2$ are continuous positive definite functions and $W$ is a continuous positive semi-definite on $\mathcal{D}$. Choose $r > 0$ and $c > 0$ such that $\bar{\mathcal{B}}(0, r) \subset \mathcal{D}$ and $c < \min_{\|x\|=r} W_1(x)$. Then for all Filippov solutions $x : [t_0, t_1] \to \mathbb{R}^n$ of (7), with $x(t_0) \in \mathbb{D} := \{x \in \bar{\mathcal{B}}(0, r) : W_2(x) \le c\}$, it holds that $t_1 = \infty$, $x(t) \in \mathbb{D}$, $\forall t \in [t_0, \infty)$, and $\lim_{t \to \infty} W(x(t)) = 0$.*

*Proof of Theorem 1.* We re-write first the condition of Assumption 2. Note that $p_\mathrm{d}(t)$ and its derivatives are bounded functions of time. Moreover, since $e = x - p_\mathrm{d}$, $\dot{e} = \dot{x} - \dot{p}_\mathrm{d}$, it holds that $\|\bar{x}\| \le \|e\| + \|\dot{e}\| + \|p_\mathrm{d}\| + \|\dot{p}_\mathrm{d}\|$ and $\dot{v}_\mathrm{d} = \ddot{p}_\mathrm{d} - k_1 \dot{e}$, as well as $e_v = \dot{x} - v_\mathrm{d} = \dot{e} + k_1 e$ implying $\dot{e} = e_v - k_1 e$. Therefore, in view of (2), one can find positive constants $d_1, d_2, D$ such that

$$\frac{1}{\underline{g}}\|\underline{g}e + f(\bar{x}, t) + g(\bar{x}, t)u_\mathrm{nn}(\bar{x}, t) - \dot{v}_\mathrm{d}\| \le d_1\|e\| + d_2\|e_v\| + D, \tag{8}$$

for all $\bar{x} \in \mathbb{R}^{2n}$, $t \ge 0$, where $\underline{g}$ is the minimum eigenvalue of $g(\bar{x}, t)$, which is positive owing to the positive definitiveness of $g(\cdot)$. Inequality (8) will be used later in the proof.

---

[4]See (Fischer et al. (2013)) for a definition of regular functions.

Let now a constant $\alpha$ such that $\frac{d_1\alpha}{2} < k_1$. As will be clarified later, the adaptation variables $\hat{\ell}_1$, $\hat{\ell}_2$ aim to approximate the constants $\frac{d_1}{2\alpha} + d_2$ and $D$, respectively. Therefore, let $\ell_1 := \frac{d_1}{2\alpha} + d_2$, $\ell_2 := D$, and the respective error terms $\tilde{\ell}_1 := \hat{\ell}_1 - \ell_1$, $\tilde{\ell}_2 := \hat{\ell}_2 - \ell_2$, as well as the overall state $\tilde{x} := [e^\top, e_v^\top, \tilde{\ell}_1, \tilde{\ell}_2]^\top \in \mathbb{R}^{2n+2}$. Since the control policy is discontinuous, we use the notion of Filippov solutions. The Filippov regularization of $u$ is $\mathsf{K}[u] = u_{\text{nn}}(x,t) - k_2 e_v - \hat{\ell}_1 e_v - \hat{\ell}_2 \hat{\mathsf{E}}_v$, where $\hat{\mathsf{E}}_v := \frac{e_v}{\|e_v\|}$ if $e_v \neq 0$ and $\hat{\mathsf{E}}_v \in (-1,1)$ otherwise. Note that, in any case, it holds that $e_v^\top \hat{\mathsf{E}}_v = \|e_v\|$.

Let now the continuously differentiable function

$$V(\tilde{x}) := \frac{1}{2}\|e\|^2 + \frac{1}{2g}\|e_v\|^2 + \sum_{i\in\{1,2\}} \frac{1}{2k_{\ell_i}} \tilde{\ell}_i$$

which satisfies $W_1(\tilde{x}) \leq V(\tilde{x}) \leq W_2(\tilde{x})$ for $W_1(\tilde{x}) := \min\{\frac{1}{2}, \frac{1}{2g}, \frac{1}{2k_{\ell_1}}, \frac{1}{2k_{\ell_2}}\}\|\tilde{x}\|^2$, $W_2(\tilde{x}) := \max\{\frac{1}{2}, \frac{1}{2g}, \frac{1}{2k_{\ell_1}}, \frac{1}{2k_{\ell_2}}\}\|\tilde{x}\|^2$. According to Lemma 1, $\dot{V}(\tilde{x}) \overset{a.e.}{\in} \dot{\tilde{V}}(\tilde{x})$, with $\dot{\tilde{V}} := \bigcap_{\xi\in\partial V(\tilde{x})} \mathsf{K}[\dot{\tilde{x}}]$. Since $V(\tilde{x})$ is continuously differentiable, its generalized gradient reduces to the standard gradient and thus it holds that $\dot{\tilde{V}}(\tilde{x}) = \nabla V^\top \mathsf{K}[\dot{\tilde{x}}]$, where $\nabla V = [e^\top, \frac{1}{g}e_v^\top, \frac{1}{k_{\ell_1}}\tilde{\ell}_1, \frac{1}{k_{\ell_2}}\tilde{\ell}_2]^\top$. By differentiating $V$ and using (3), one obtains

$$\dot{V} \subset \widetilde{W}_s := e^\top(\dot{x} - \dot{p}_{\text{d}}) + \frac{1}{g}e_v^\top\big(f(\bar{x},t) + g(\bar{x},t)u - \dot{v}_{\text{d}}\big) + \frac{1}{k_{\ell_1}}\tilde{\ell}_1\dot{\hat{\ell}}_1 + \frac{1}{k_{\ell_2}}\tilde{\ell}_2\dot{\hat{\ell}}_2$$

and by using $\dot{x} = e_v + v_{\text{d}}$ and substituting the control policy (3), (4), and inequality (8),

$$\widetilde{W}_s := - k_1\|e\|^2 + \frac{1}{g}e_v^\top\big(ge + f(\bar{x},t) + g(\bar{x},t)u_{\text{nn}}(\bar{x},t) - \dot{v}_{\text{d}}\big) - \frac{1}{g}e_v^\top g(\bar{x},t)\big(k_2 e_v + \hat{\ell}_1 e_v + \hat{\ell}_2 \hat{\mathsf{E}}_v\big)$$

$$+ \tilde{\ell}_1\|e_v\|^2 + \tilde{\ell}_2\|e_v\|$$

$$\leq - k_1\|e\|^2 + d_1\|e_v\|\|e\| + d_2\|e_v\|^2 + D\|e_v\| - \frac{1}{g}e_v^\top g(\bar{x},t)\big(k_2 e_v + \hat{\ell}_1 e_v + \hat{\ell}_2 \hat{\mathsf{E}}_v\big)$$

$$+ \tilde{\ell}_1\|e_v\|^2 + \tilde{\ell}_2\|e_v\|$$

Note from (4) and the fact that $\hat{\ell}_1(0) > 0$, $\hat{\ell}_2(0) > 0$ that $\hat{\ell}_1(t) > 0$ and $\hat{\ell}_2(t)) > 0$ for all $t \geq 0$. Moreover, recall that $g(\bar{x},t)$ is positive definite for all $\bar{x} \in \mathbb{R}^{2n}$, $t \geq 0$, with $g$ being its minimum (and positive) eigenvalue. Therefore, we conclude that $\widetilde{W}_s$ becomes

$$\widetilde{W}_s \leq - k_1\|e\|^2 + d_1\|e_v\|\|e\| + d_2\|e_v\|^2 + D\|e_v\| - (k_2 + \hat{\ell}_1)\|e_v\|^2 - \hat{\ell}_2\|e_v\| + \tilde{\ell}_1\|e_v\|^2 + \tilde{\ell}_2\|e_v\|$$

By incorporating the term $\alpha$ in the term $d_1\|e_v\|\|e\|$, we obtain $d_1\|e_v\|\|e\| = d_1\alpha\frac{\|e_v\|}{\alpha}\|e\|$ and by using the property $ab \leq \frac{1}{2}a^2 + \frac{1}{2}b^2$ for any constants $a,b$, we obtain $d_1\alpha\frac{\|e_v\|}{\alpha}\|e\| \leq \frac{d_1\alpha}{2}\|e\|^2 + \frac{d_1}{2\alpha}\|e_v\|^2$. Therefore, $\widetilde{W}_s$ becomes

$$\widetilde{W}_s \leq -\left(k_1 - \frac{d_1\alpha}{2}\right)\|e\|^2 + \left(\frac{d_1}{2\alpha} + d_2\right)\|e_v\|^2 + D\|e_v\| - (k_2 + \hat{\ell}_1)\|e_v\|^2 - \hat{\ell}_2\|e_v\| + \tilde{\ell}_1\|e_v\|^2 + \tilde{\ell}_2\|e_v\|$$

$$= -\left(k_1 - \frac{d_1\alpha}{2}\right)\|e\|^2 + \ell_1\|e_v\|^2 + \ell_2\|e_v\| - (k_2 + \hat{\ell}_1)\|e_v\|^2 - \hat{\ell}_2\|e_v\| + \tilde{\ell}_1\|e_v\|^2 + \tilde{\ell}_2\|e_v\|$$

$$= -\left(k_1 - \frac{d_1\alpha}{2}\right)\|e\|^2 - k_2\|e_v\|^2 =: -Q(\tilde{x})$$

Therefore, it holds that $\zeta \leq -Q(\tilde{x})$, for all $\zeta \in \dot{\tilde{V}}$, with $Q$ being a continuous and positive semi-definite function in $\mathbb{R}^{2n+2}$, since $k_1 - \frac{d_1\alpha}{2} > 0$. Choose now any finite $r > 0$ and let $c < \min_{\|\tilde{x}\|=r} W_1(\tilde{x})$. Hence, all the conditions of Corollary 1 are satisfied and hence, all Filippov solutions starting from $\tilde{x}(0) \in \Omega_f := \{\tilde{x} \in \mathcal{B}(0,r) : \widetilde{W}_2(\tilde{x}) \leq c\}$ are bounded and remain in $\Omega_f$, satisfying $\lim_{t\to\infty} Q(\tilde{x}(t)) = 0$. Note that $r$, and hence $c$, can be arbitrarily large allowing and finite initial condition $\tilde{x}(0)$. Moreover, the boundedness of $\tilde{x}$ implies the boundedness of $e$, $\dot{e}$, $e_v$, $\tilde{\ell}_1$, and $\tilde{\ell}_2$, and hence of $\hat{\ell}_1(t)$ and $\hat{\ell}_2(t)$, for all $t \in \mathbb{R}_{\geq 0}$. In view of (5), we finally conclude the boundedness of $u(\cdot)$, $\dot{\hat{\ell}}_1$, and $\dot{\hat{\ell}}_2$, for all $t \in \mathbb{R}_{\geq 0}$, leading to the conclusion of the proof.

$\square$

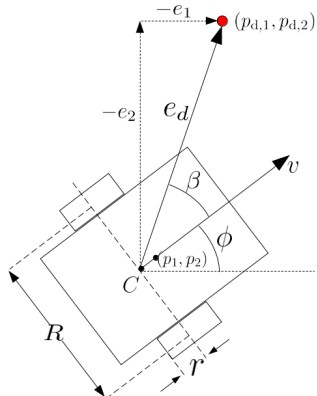

Figure 6: A unicycle vehicle.

## B  APPENDIX

**Extension to non-holonomic unicycle dynamics**

As mentioned in Section 1, the dynamics 1 do not represent all kinds of systems, with one particular example being when non-holonomic constraints are present. In such cases, the control law design (4) and Theorem 1 no longer hold. In this section, we extend the control policy to account for unicycle vehicles subject to first-order non-holonomic constraints. More specifically, we consider the dynamics

$$\dot{p}_1 = v \cos\phi, \ \ \dot{p}_2 = v \sin\phi, \ \ \dot{\phi} = \omega \tag{9a}$$

$$M\ddot{\theta} = u + f_\theta(\bar{x}, t) \tag{9b}$$

where $x = [p_1, p_2, \phi]^\top \in \mathbb{R}^3$ are the unicycle's position and orientation, $(v, \omega)$ are its linear and angular velocity (see Fig. 6), $\theta := [\theta_R, \theta_L]^\top \in \mathbb{R}^2$ are its wheel's angular positions, and $u = [u_R, u_L]^\top \in \mathbb{R}^2$ are the wheel's torques, representing the control input. The unicycle vehicle is subject to the non-holonomic constraint $\dot{p}_1 \sin\phi - \dot{p}_2 \cos\phi = 0$, which implies that the vehicle cannot move laterally. Additionally, $M \in \mathbb{R}^{2\times2}$ is the vehicle's inertia matrix, which is symmetric and positive definite, and $f_\theta(\cdot)$ is a function representing friction and external disturbances. The velocities satisfy the relations $v = \frac{r}{2}(\dot{\theta}_R + \dot{\theta}_L)$, $\omega = \frac{r}{2R}(\dot{\theta}_R - \dot{\theta}_L))$, where $r$ and $R$ are the wheels' radius and axle length, respectively. The terms $r$, $R$, $M$, and $f_\theta(\cdot)$ are considered to be *completely unknown*. As before, the goal is for the vehicle's position $p := [p_1, p_2]^\top$ to track a desired trajectory $p_d = [p_{d,1}, p_{d,2}]^\top \in \mathbb{R}^2$. Towards that end, we define the error variables $e_1 := p_1 - p_{d,1}$, $e_2 := p_2 - p_{d,2}$, $e_d := \|p - p_d\|$, as well as the angle $\beta$ measured from the the longitudinal axis of the vehicle, i.e., the unicycle's direction vector $[\cos\phi, \sin\phi]$, to the error vector $-[e_1, e_2]$ (see Fig. 6). The angle $\beta$ can be derived by using the cross product between the aforementioned vectors, i.e., $e_d \sin(\beta) = [\cos\phi, \sin\phi] \times [-e_1, -e_2] = e_1 \sin\phi - e_2 \cos\phi$. The purpose of the control design, illustrated next, is to drive $e_d$ and $\beta$ to zero. By differentiating the latter and using (9) as well as the relations $e_1 = -e_d \cos(\phi + \beta)$, $e_2 = -e_d \sin(\phi + \beta)$ (see Fig. 6), we derive

$$\dot{e}_d = -v \cos\beta + \dot{p}_{d,1} \cos(\phi + \beta) + \dot{p}_{d,2} \sin(\phi + \beta) \tag{10a}$$

$$\dot{\beta} = -\omega + \frac{\sin\beta}{e_d}v - \frac{\dot{p}_{d,1}}{e_d}\sin(\phi + \beta) + \frac{\dot{p}_{d,2}}{e_d}\cos(\phi + \beta) \tag{10b}$$

In view of (10), we set reference signals for the vehicle's velocity as

$$v_d := \frac{1}{\cos(\beta)}(\dot{p}_{d,1}\cos(\beta + \phi) + \dot{p}_{d,2}\sin(\beta + \phi) + k_d e_d) \tag{11a}$$

$$\omega_d := -\frac{\sin(\phi)\dot{p}_{d,1}}{\cos(\beta)e_d} + \frac{\cos(\phi)\dot{p}_{d,2}}{\cos(\beta)e_d} + k_d \tan\beta + k_\beta \beta \tag{11b}$$

where $k_d, k_\beta$ are positive gains, aiming to create exponentially stable subsystems via the terms $k_d e_d$ and $k_\beta \beta$. We define next the respective velocity errors $e_v := v - v_d$, $e_\omega := \omega - \omega_d$ and design the

adaptive and neural-network-based control input as $u(\bar{x}, \hat{d}, t) := [\frac{u_S + u_D}{2}, \frac{u_S - u_D}{2}]^\top + u_{\text{nn}}(\bar{x}, t)$, with

$$u_S := \hat{\ell}_v \dot{v}_{\text{d}} - (k_v + \hat{\ell}_1)e_v - \hat{\ell}_2 \hat{e}_v + e_d \cos \beta - \beta \frac{\sin \beta}{e_d} \tag{12a}$$

$$u_D := \hat{\ell}_\omega \dot{\omega}_{\text{d}} - (k_\omega + \hat{\ell}_1)e_\omega - \hat{\ell}_2 \hat{e}_\omega + \beta \tag{12b}$$

$$\dot{\hat{\ell}}_v := -k_v e_v \dot{v}_{\text{d}}, \qquad \dot{\hat{\ell}}_\omega := -k_\omega e_\omega \dot{\omega}_{\text{d}} \tag{12c}$$

$$\dot{\hat{\ell}}_1 := k_1(e_v^2 + e_\omega^2) \qquad \dot{\hat{\ell}}_2 := k_2(|e_v| + |e_\omega|) \tag{12d}$$

where $\hat{\ell}_v, \hat{\ell}_\omega, \hat{\ell}_i$ are adaptation variables (similar to (4)), with $\hat{\ell}_v(0) > 0, \hat{\ell}_\omega(0) > 0$ and $k_v, k_\omega, k_i,$ are positive gains, $i \in \{1, 2\}$; $\hat{e}_a$, with $a \in \{v, \omega\}$, is defined as $\hat{e}_a = \frac{e_a}{|e_a|}$ if $e_a \neq 0$ and $\hat{e}_a = 0$ otherwise. We now re-state Assumption 2 to apply for the unicycle analysis as follows.

**Assumption 3.** *The output $u_{\text{nn}}(\bar{x}, t)$ of the trained neural network satisfies $\|u_{\text{nn}}(\bar{x}, t) + f_\theta(\bar{x}, t)\| \leq d\|\bar{x}\| + B$, for positive, unknown constants $d, B$.*

Similar to assumption 2, assumption 3 is merely a growth-boundedness condition by the unknown constants $d$ and $B$. The stability of the proposed scheme is provided in the next corollary.

**Corollary 2.** *Let the unicycle system (9) and let an open-loop trajectory $p_{\text{d}} : \mathbb{R}_{\geq 0} \to \mathbb{R}^2$. Assume that $\beta(t) \in (-\bar{\beta}, \bar{\beta})$, $|\dot{p}_{\text{d},1} \sin \phi - \dot{p}_{\text{d},2} \cos \phi| < e_d \alpha_1$, $\sin \beta < e_d \alpha_2$ for positive constants $\bar{\beta} < \frac{\pi}{2}$, $\alpha_1, \alpha_2$ and all $t \geq 0$. Under Assumption 3, the control policy (12) guarantees $\lim_{t \to \infty}(e_d(t), \beta(t), e_v(t), e_\omega(t)) = 0$, and the boundedness of all closed-loop signals.*

*Proof of Corollary 2.* The derivatives of $e_d, \beta$ in (10) can be written as

$$\begin{bmatrix} \dot{e}_d \\ \dot{\beta} \end{bmatrix} = G \begin{bmatrix} v \\ \omega \end{bmatrix} + \begin{bmatrix} \dot{p}_{\text{d},1} \cos(\phi + \beta) + \dot{p}_{\text{d},2} \sin(\phi + \beta) \\ \frac{\dot{p}_{\text{d},2}}{e_d} \cos(\phi + \beta) - \frac{\dot{p}_{\text{d},1}}{e_d} \sin(\phi + \beta) \end{bmatrix}$$

where

$$G := \begin{bmatrix} -\cos \beta & 0 \\ \frac{\sin \beta}{e_d} & -1 \end{bmatrix}$$

It can be verified that the reference velocity signals, designed in (11), can be written as

$$\begin{bmatrix} v_{\text{d}} \\ \omega_{\text{d}} \end{bmatrix} = G^{-1} \begin{bmatrix} -\dot{p}_{\text{d},1} \cos(\phi + \beta) - \dot{p}_{\text{d},2} \sin(\phi + \beta) - k_d e_d \\ -\frac{\dot{p}_{\text{d},2}}{e_d} \cos(\phi + \beta) + \frac{\dot{p}_{\text{d},1}}{e_d} \sin(\phi + \beta) - k_\beta \beta \end{bmatrix}$$

and therefore, by using the relations $v = e_v + v_{\text{d}}$ and $\omega = e_\omega + \omega_{\text{d}}$, $\dot{e}_d$ and $\dot{\beta}$ can be written as

$$\begin{bmatrix} \dot{e}_d \\ \dot{\beta} \end{bmatrix} = \begin{bmatrix} -k_d e_d \\ -k_\beta \beta \end{bmatrix} + G \begin{bmatrix} e_v \\ e_\omega \end{bmatrix} = \begin{bmatrix} -k_d e_d \\ -k_\beta \beta \end{bmatrix} + \begin{bmatrix} -e_v \cos \beta \\ e_v \frac{\sin \beta}{e_d} - e_\omega \end{bmatrix} \tag{13}$$

The control design follows the back-stepping methodology (Krstic et al. (1995)). Let, therefore, the function $V_1 := \frac{1}{2}(e_d^2 + \beta^2)$, which, after time differentiation and use of (13), yields

$$\dot{V}_1 = -k_d e_d^2 - k_\beta \beta^2 - e_d e_v \cos \beta + \beta e_v \frac{\sin \beta}{e_d} - \beta e_\omega \tag{14}$$

We will cancel the two last terms of $\dot{V}_1$ using the control input (12).

First, we note that the inertia matrix of the system $M$, appearing in the unicycle dynamics (9), has the form (Ivanjko et al. (2010)) $M = \begin{bmatrix} M_1 & M_2 \\ M_2 & M_1 \end{bmatrix}$ with $M_1 := \frac{mr^2}{4} + \frac{(I_C + md^2)r^2}{4R^2} + I_0$, $M_2 := \frac{mr^2}{4} - \frac{(I_C + md^2)r^2}{4R^2}$, and where $I_C$ is the moment of inertia of the vehicle with respect to point C (see Fig. 6), $I_0$ is the moment of inertia of the the wheels, and $d$ is the distance of the between

point C and the vehicle's center of mass $(p_1, p_2)$. Therefore, the second part of the unicycle dynamics (9) can be written as

$$M_1\ddot{\theta}_R + M_2\ddot{\theta}_L = u_R + f_{\theta,R}(\bar{x}, t)$$
$$M_2\ddot{\theta}_R + M_1\ddot{\theta}_L = u_L + f_{\theta,L}(\bar{x}, t)$$

with $f_{\theta,R}$ and $f_{\theta,L}$ denoting the elements of $f_\theta$. By summing and subtracting the aforementioned equations, we obtain

$$(M_1 + M_2)(\ddot{\theta}_R + \ddot{\theta}_L) = u_R + u_L + f_{\theta,R}(\bar{x}, t) + f_{\theta,L}(\bar{x}, t) \tag{15a}$$
$$(M_1 - M_2)(\ddot{\theta}_R - \ddot{\theta}_L) = u_R - u_L + f_{\theta,R}(\bar{x}, t) - f_{\theta,L}(\bar{x}, t) \tag{15b}$$

From the definition of $e_v$ and $e_\omega$, it holds that $\dot{e}_v = \dot{v} - \dot{v}_{\mathrm{d}}$, $\dot{e}_\omega = \dot{\omega} - \dot{\omega}_{\mathrm{d}}$, which, by using the relations $v = \frac{r}{2}(\dot{\theta}_R + \dot{\theta}_L)$, $\omega = \frac{r}{2R}(\dot{\theta}_R - \dot{\theta}_L)$ and (15), becomes

$$\dot{e}_v = \frac{r}{2(M_1 + M_2)}\left(u_R + u_L + f_{\theta,R}(\bar{x}, t) + f_{\theta,L}(\bar{x}, t)\right) - \dot{v}_{\mathrm{d}} \tag{16a}$$
$$\dot{e}_\omega = \frac{r}{2R(M_1 - M_2)}\left(u_R - u_L + f_{\theta,R}(\bar{x}, t) - f_{\theta,L}(\bar{x}, t)\right) - \dot{\omega}_{\mathrm{d}} \tag{16b}$$

Define now $\ell_v := 2\frac{M_1 + M_2}{r}$, $\ell_\omega := 2R\frac{M_1 - M_2}{r}$. The adaptation terms $\hat{\ell}_v$, and $\hat{\ell}_\omega$, used in the control mechanism (12), aim to approximate $\ell_v$ and $\ell_\omega$, respectively. Note that $\hat{\ell}_v$ and $\hat{\ell}_\omega$ are positive, and we define the errors $\widetilde{\ell}_v := \hat{\ell}_v - \ell_v$, $\widetilde{\ell}_\omega := \hat{\ell}_\omega - \ell_\omega$.

We re-write next the condition of Assumption 3. It holds that $\|p\| = \sqrt{p_1^2 + p_2^2} \le 2e_d + |p_{\mathrm{d},1}| + |p_{\mathrm{d},2}|$. Moreover, in view of (11) as well as the fact that $\beta \in (-\bar{\beta}, \bar{\beta}) \subset (-\frac{\pi}{2}, \frac{\pi}{2})$, where $\bar{\beta}$ is the bound of $\beta$, stated in Corollary 2, it holds that $|v| \le |e_v| + |v_{\mathrm{d}}| \le |e_v| + \frac{1}{\cos(\bar{\beta})}(|\dot{p}_{\mathrm{d},1}| + |\dot{p}_{\mathrm{d},2}| + k_d e_d)$, $|\omega| \le |e_\omega| + |\omega_{\mathrm{d}}| \le |e_\omega| + \frac{1}{\cos\bar{\beta}}(\alpha_1 + k_d) + k_\beta|\beta|$, where we also use the fact that $\frac{|\dot{p}_{\mathrm{d},1}\sin\phi - \dot{p}_{\mathrm{d},2}\cos\phi|}{e_d} \le \alpha_1$. By also recalling that $\phi$ moves on the unit circle, we conclude that Assumption 3 becomes

$$\|f_\theta(\bar{x}, t) + u_{\mathrm{nn}}(\bar{x}, t)\| \le d_d e_d + d_\beta|\beta| + d|e_v| + d|e_\omega| + D, \tag{17}$$

with $d_d := d(2 + \frac{k_d}{\cos\bar{\beta}})$, $d_\beta := dk_\beta$, and $D := d(2\bar{p}_{\mathrm{d}} + 2\pi + \frac{1}{\cos\bar{\beta}}(\alpha_1 + 2\bar{v}_{\mathrm{d}} + 1))$, where $\bar{p}_{\mathrm{d}}$ and $\bar{v}_{\mathrm{d}}$ are the upper bounds of $p_{\mathrm{d},i}$, $v_{\mathrm{d},i}$, respectively, $i \in \{1, 2\}$.

Let now positive constants $\gamma_d$ and $\gamma_\beta$ such that $k_d > 2d_d\gamma_d$ and $k_\beta > 2d_\beta\gamma_\beta$, and define

$$\ell_1 := \frac{d_d}{\gamma_d} + \frac{d_\beta}{\gamma_\beta} + 4d \tag{18}$$

which is the constant to be approximated by $\hat{\ell}_1$; its derivation will be clarified later. Let also $\ell_2 := 2D$, which we aim to approximate with the adaptation variable $\hat{\ell}_2$. We define hence the respective errors $\widetilde{\ell}_1 := \hat{\ell}_1 - \ell_1$ and $\widetilde{\ell}_2 := \hat{\ell}_2 - \ell_2$ and define the overall vector $\widetilde{x} := [e_d, \beta, e_v, e_\omega, \widetilde{\ell}_v, \widetilde{\ell}_\omega, \widetilde{\ell}_1, \widetilde{\ell}_2]^\top \in \mathbb{R}^8$. Since the control mechanism is discontinuous, we use again the notion of Filippov solutions. The Filippov regularization $\mathsf{K}[u]$ of $u$ differs from $u$ only in the terms $\hat{e}_v$ and $\hat{e}_\omega$, which are replaced by $\hat{\mathsf{E}}_v$, defined as $\hat{\mathsf{E}}_v := \frac{e_v}{|e_v|}$ if $e_v \ne 0$, and $\hat{\mathsf{E}}_v \in (-1, 1)$ otherwise, and $\hat{\mathsf{E}}_\omega$, defined as $\hat{\mathsf{E}}_\omega := \frac{e_\omega}{|e_\omega|}$ if $e_\omega \ne 0$, and $\hat{\mathsf{E}}_\omega \in (-1, 1)$, respectively.

Consider now the continuously differentiable and positive definite function

$$V(\bar{x}) := V_1 + \frac{\ell_v}{2}e_v^2 + \frac{\ell_\omega}{2}e_\omega^2 + \frac{1}{2k_v}\widetilde{\ell}_v^2 + \frac{1}{2k_\omega}\widetilde{\ell}_\omega^2 + \sum_{i\in\{1,2\}}\frac{1}{2k_i}\widetilde{\ell}_i^2$$

which satisfies $W_1(\widetilde{x}) \le V(\widetilde{x}) \le W_2(\widetilde{x})$ for $W_1(\widetilde{x}) := \min\{\frac{1}{2}, \frac{\ell_v}{2}, \frac{\ell_\omega}{2}, \frac{1}{2k_v}, \frac{1}{2k_\omega}, \frac{1}{2k_1}, \frac{1}{2k_2}\}\|\widetilde{x}\|^2$, $W_2(\widetilde{x}) := \max\{\frac{1}{2}, \frac{\ell_v}{2}, \frac{\ell_\omega}{2}, \frac{1}{2k_v}, \frac{1}{2k_\omega}, \frac{1}{2k_1}, \frac{1}{2k_2}\}\|\widetilde{x}\|^2$. According to Lemma 1, $\dot{V}(\widetilde{x}) \overset{a.e.}{\in} \dot{\widetilde{V}}(\widetilde{x})$,

with $\dot{\widetilde{V}} := \bigcap_{\xi \in \partial V(\widetilde{x})} \mathsf{K}[\dot{\widetilde{x}}]$. Since $V(\widetilde{x})$ is continuously differentiable, its generalized gradient reduces to the standard gradient and thus it holds that $\dot{\widetilde{V}}(\widetilde{x}) = \nabla V^\top \mathsf{K}[\dot{\widetilde{x}}]$, where $\nabla V = [e_d, \beta, \ell_v e_v, \ell_\omega e_\omega, \frac{1}{k_v}\widetilde{\ell}_v, \frac{1}{k_\omega}\widetilde{\ell}_\omega, \frac{1}{k_1}\widetilde{\ell}_1, \frac{1}{k_2}\widetilde{\ell}_2]^\top$. Therefore, by differentiating $V$ and employing (14) and (15), we obtain

$$\dot{V} \subset \widetilde{W} := - k_d e_d^2 - k_\beta \beta^2 - e_d e_v \cos\beta + \beta e_v \frac{\sin\beta}{e_d} - \beta e_\omega + e_v\big(u_R + u_L + f_{\theta,R} + f_{\theta,L} - \ell_v \dot{v}_{\mathrm d}\big)$$

$$+ e_\omega\big(u_R - u_L + f_{\theta,R} - f_{\theta,L} - \ell_\omega \dot{\omega}_{\mathrm d}\big) + \frac{1}{k_v}\widetilde{\ell}_v \dot{\widetilde{\ell}}_v + \frac{1}{k_\omega}\widetilde{\ell}_\omega \dot{\widetilde{\ell}}_\omega + \sum_{i \in \{1,2\}} \frac{1}{k_i}\widetilde{\ell}_i \dot{\widetilde{\ell}}_i$$

By substituting the control and adaptation policy (12), $\widetilde{W}$ becomes

$$\widetilde{W} = - k_d e_d^2 - k_\beta \beta^2 - k_v e_v^2 - k_\beta e_\omega^2 + e_v \dot{v}_{\mathrm d}\widetilde{\ell}_v + e_\omega \dot{\omega}_{\mathrm d}\widetilde{\ell}_\omega - \hat{\ell}_1(e_v^2 + e_\omega^2) - \hat{\ell}_2(|e_v| + |e_\omega|)$$

$$+ e_v(u_{\mathrm{nn},R} + u_{\mathrm{nn},L} + f_{\theta,R} + f_{\theta,L}) + e_\omega(u_{\mathrm{nn},R} - u_{\mathrm{nn},L} + f_{\theta,R} - f_{\theta,L})$$

$$- \widetilde{\ell}_v e_v \dot{v}_{\mathrm d} - \widetilde{\ell}_\omega e_\omega \dot{\omega}_{\mathrm d} + \widetilde{\ell}_1(e_v^2 + e_\omega^2) + \widetilde{\ell}_2(|e_v| + |e_\omega|)$$

where $u_{R,\mathrm{nn}}$ and $u_{L,\mathrm{nn}}$ are the elements of $u_{\mathrm{nn}}$, i.e., $u_{\mathrm{nn}} = [u_{R,\mathrm{nn}}, u_{L,\mathrm{nn}}]^\top$. It is easy to conclude that $|u_{\mathrm{nn},R} + u_{\mathrm{nn},L} + f_{\theta,R} + f_{\theta,L}| \le |u_{\mathrm{nn},R} + f_{\theta,R}| + |u_{\mathrm{nn},L} + f_{\theta,L}| \le 2\|u_{\mathrm{nn}} + f_\theta\|$ and $|u_{\mathrm{nn},R} - u_{\mathrm{nn},L} + f_{\theta,R} - f_{\theta,L}| \le |u_{\mathrm{nn},R} + f_{\theta,R}| + |u_{\mathrm{nn},L} + f_{\theta,L}| \le 2\|u_{\mathrm{nn}} + f_\theta\|$ and hence, in view of (17),

$$\left.\begin{array}{c} |u_{\mathrm{nn},R} + u_{\mathrm{nn},L} + f_{\theta,R} + f_{\theta,L}| \\ |u_{\mathrm{nn},R} - u_{\mathrm{nn},L} + f_{\theta,R} - f_{\theta,L}| \end{array}\right\} \le \bar{D} := 2d_d e_d + 2d_\beta|\beta| + 2d|e_v| + 2d|e_\omega| + 2D$$

Therefore, $\widetilde{W}$ becomes

$$\widetilde{W} = - k_d e_d^2 - k_\beta \beta^2 - k_v e_v^2 - k_\beta e_\omega^2 - \hat{\ell}_1(e_v^2 + e_\omega^2) - \hat{\ell}_2(|e_v| + |e_\omega|) + \bar{D}(|e_v| + |e_\omega|)$$

$$+ \widetilde{\ell}_1(e_v^2 + e_\omega^2) + \widetilde{\ell}_2(|e_v| + |e_\omega|)$$

The term $\bar{D}(|e_v| + |e_\omega|)$ yields

$$\bar{D}(|e_v| + |e_\omega|) = 2d_d e_d|e_v| + 2d_\beta|\beta||e_v| + 2de_v^2 + 4d|e_v||e_\omega| + 2D|e_v| + 2d_d e_d|e_\omega|$$

$$+ 2d_\beta|\beta||e_\omega| + 2de_\omega^2 + 2D|e_\omega|$$

By setting $2d_d e_d|e_v| = 2d_d\gamma_d e_d \frac{|e_v|}{\gamma_d}$, $2d_d e_d|e_\omega| = 2d_d\gamma_d e_d\frac{|e_\omega|}{\gamma_d}$, and $2d_\beta|\beta||e_v| = 2d_\beta\gamma_\beta|\beta|\frac{|e_v|}{\gamma_\beta}$, $2d_\beta|\beta||e_\omega| = 2d_\beta\gamma_\beta|\beta|\frac{|e_\omega|}{\gamma_\beta}$ and completing the squares, one obtains

$$\bar{D}(|e_v| + |e_\omega|) \le 2d_d\gamma_d e_d^2 + 2d_\beta\gamma_\beta\beta^2 + \left(\frac{d_d}{\gamma_d} + \frac{d_\beta}{\gamma_\beta} + 4d\right)(e_v^2 + e_\omega^2) + 2D(|e_v| + |e_\omega|)$$

$$= 2d_d\gamma_d e_d^2 + 2d_\beta\gamma_\beta\beta^2 + \ell_1(e_v^2 + e_\omega^2) + \ell_2(|e_v| + |e_\omega|)$$

Hence, $\widetilde{W}$ becomes

$$\widetilde{W} \le - (k_d - 2d_d\gamma_d)e_d^2 - (k_\beta - 2d_\beta\gamma_\beta)\beta^2 - k_v e_v^2 - k_\beta e_\omega^2 - \hat{\ell}_1(e_v^2 + e_\omega^2) - \hat{\ell}_2(|e_v| + |e_\omega|)$$

$$+ \ell_1(e_v^2 + e_\omega^2) + \ell_2(|e_v| + |e_\omega|) + \widetilde{\ell}_1(e_v^2 + e_\omega^2) + \widetilde{\ell}_2(|e_v| + |e_\omega|)$$

$$= - (k_d - 2d_d\gamma_d)e_d^2 - (k_\beta - 2d_\beta\gamma_\beta)\beta^2 - k_v e_v^2 - k_\beta e_\omega^2 =: Q(\widetilde{x})$$

Therefore, it holds that $\zeta \le -Q(\widetilde{x})$, for all $\zeta \in \dot{\widetilde{V}}$, with $Q$ being a continuous and positive semi-definite function in $\mathbb{R}^8$, since $k_d - 2d_d\gamma_d > 0$, $k_\beta - 2d_\beta\gamma_\beta > 0$. Choose now any finite $r > 0$ and let $c < \min_{\|\widetilde{x}\|=r} W_1(\widetilde{x})$. Hence, all the conditions of Corollary 1 are satisfied and hence, all Filippov solutions starting from $\widetilde{x}(0) \in \Omega_f := \{\widetilde{x} \in \mathcal{B}(0,r) : \widetilde{W}_2(\widetilde{x}) \le c\}$ are bounded and remain in $\Omega_f$, satisfying $\lim_{t \to \infty} Q(\widetilde{x}(t)) = 0$.

Note that $r$, and hence $c$, can be arbitrarily large allowing and finite initial condition $\widetilde{x}(0)$. Moreover, the boundedness of $\widetilde{x}$ implies the boundedness of $e$, $\dot{e}$, $e_v$, $\widetilde{\ell}_1$, and $\widetilde{\ell}_2$, and hence of $\hat{\ell}_1(t)$ and $\hat{\ell}_2(t)$, for all $t \in \mathbb{R}_{\ge 0}$. Finally, in view of the conditions $\frac{|\sin\phi \dot{p}_{\mathrm d,1} - \cos\phi \dot{p}_{\mathrm d,2}|}{e_d} < \alpha_1$ and $\frac{\sin\beta}{e_d} < \alpha_2$, one concludes the boundedness of all closed-loop signals, for all $t \ge 0$

$\square$

The assumptions $|\dot{p}_{d,1} \sin\phi - \dot{p}_{d,2} \cos\phi| < e_d\alpha_1$, $\sin\beta < e_d\alpha_2$ are imposed to avoid the singularity of $e_d = 0$; note that $\beta$ and $\omega_d$ are not defined in that case. Intuitively, they imply that $e_d$ will not be driven to zero faster than $\beta$ or $\dot{p}_{d,1} \sin\phi - \dot{p}_{d,2} \cos\phi$; the latter becomes zero when the vehicle's velocity vector $v$ aligns with the desired one $\dot{p}_d$. In the experiments, we tune the control gains according to $k_\beta \sim 10k_d$ in order to satisfy these assumptions.

# C  APPENDIX

We provide here more information on the experimental results of Section 4. All the results were obtained on MATLAB environment using the ODE simulator. We performed the training of the neural networks in with the pytorch library in Python environment. The neural networks consist of $4$ fully connected layers of $512$ neurons; each layer is followed by a batch normalization module and a ReLU activation function. For the training we use the adam optimizer and the mean-square-error loss function. In all cases we choose a batch size of $256$, and we train until a desirable average (per batch) loss of the order of $10^{-4}$ is achieved. All the values of the data used for the training were normalized in $[0, 1]$.

Regarding the robotic-arm task, the points of interest are set as $T_1 = [-0.15, -0.475, 0.675, \frac{\pi}{2}, 0, 0]^\top$, $T_2 = [-0.6, 0, 2.5, 0, -\frac{\pi}{2}, -\frac{\pi}{2}]^\top$, $T_3 = [-0.025, 0.595, 0.6, -\frac{\pi}{2}, 0, \pi]^\top$, and $T_4 = [-0.525, -0.55, 0.28, \pi, 0, -\frac{\pi}{2}]^\top$, and the corresponding joint-angle vectors as $c_1 = [-0.07, -1.05, 0.45, 2.3, 1.37, -1.33]^\top$, $c_2 = [1.28, 0.35, 1.75, 0.03, 0.1, -1.22]^\top$, $c_3 = [-0.08, 0.85, -0.23, 2.58, 2.09, -2.36]^\top$, $c_4 = [-0.7, -0.76, -1.05, -0.05, -3.08, 2.37]^\top$ (radians).

We set a nominal value for the time intervals as $I_i = [0, 20]$ (seconds), and we create 150 problem instances by varying the following attributes: firstly, we add uniformly random offsets in $[-0.3, 0.3]$ (radians) to the elements of all $c_i$, and in $[-2, 2]$ (seconds) to the right end-points of the intervals $I_i$; secondly, we add random offsets to the dynamic parameters of the robot (masses and moments of inertia of the robot's links and actuators) and we set a different friction and disturbance term $d(\cdot)$, leading to a different dynamic model in (5); thirdly, we set different sequences of visits to the points $c_i$, $i \in \{1, \ldots, 4\}$, as dictated by $\phi$, i.e., one trajectory might correspond to the visit sequence $((x(0), 0) \to (c_1, t_{1_1}) \to (c_2, t_{1_2}) \to (c_3, t_{1_3}) \to (c_4, t_{1_4})$, and another to $((x(0), 0) \to (c_3, t_{1_3}) \to (c_1, t_{1_1}) \to (c_4, t_{1_4}) \to (c_2, t_{1_2})$. Finally, we add uniformly random offsets in $[-0.5, 0.5]$ to the initial position of the robot (from the first point of the sequence), and we set its initial velocity randomly in the interval $[0, 1]^6$.

Regarding the dynamics (5), we use the methodology described in (Siciliano et al. (2009)) to derive the $B$, $C$, and $g$ terms. We set nominal link masses and moments of inertia as $m = [1, 2.5, 5.7, 3.9, 2.5, 2.5, 0.7]$ (kg) and $I = [0.02, 0.04, 0.06, 0.05, 0.04, 0.04, 0.01]$ (kgm$^2$), respectively, and we add random offsets in $(-\frac{m}{2}, \frac{m}{2})$, $(-\frac{I}{2}, \frac{I}{2})$ in the created instances. Regarding the function $d()$ used in (5); we set $d(\bar{x}, t) = d_t(t) + d_f(\bar{x})$, where

$$d_t = A_t \begin{bmatrix} \sin(\eta_1 t + \varphi_1) \\ \vdots \\ \sin(\eta_6 t + \varphi_6) \end{bmatrix}, \quad d_f = B_t \dot{x} \otimes \dot{x}$$

$A_t = \text{diag}\{ A_{t_i} \}_{i \in \{1,\ldots,6\}} \in \mathbb{R}^{6 \times 6}$, $A_{t_i}$ is a random term in $(0, 2m_i)$, $\eta_i$ is a random term in $(0, 1)$, $\varphi_i$ is a random term in $(0, 2)$, $B_t \in \mathbb{R}^{6 \times 36}$ is a random matrix taking values in $(0, 2m_i)$, and $\otimes$ denotes the Kronecker product. We chose the control gains of the control policy (4) as $k_1 = 1, k_2 = 10$, and $k_{\ell_1} = k_{\ell_2} = 10$.

Further experimental results are depicted in Figs. 7-11; Fig. 7a depicts the mean and standard deviation of $\|e_v(t)\|$ of the proposed control policy for the 50 test instances, whereas Figs. 10a depicts the control input that results from the control policy (4) as well as the the neural-network output. Note that the control input converges to the neural-network output, i.e., $\lim_{t \to \infty}(u(t) - u_{nn}(t)) = 0$, which can be also verified by (4) and the fact that $\lim_{t \to \infty} e_v(t) = 0$. Fig. 11a depicts the mean and standard deviation of the adaptation signals $\hat{\ell}_1(t)$, $\hat{\ell}_2(t)$ for the 50 test instances. Finally, Fig. 9 shows timestamps of one of the test trajectories, illustrating the visit of the robot end-effector to the points of interest at the pre-specified time stamps.

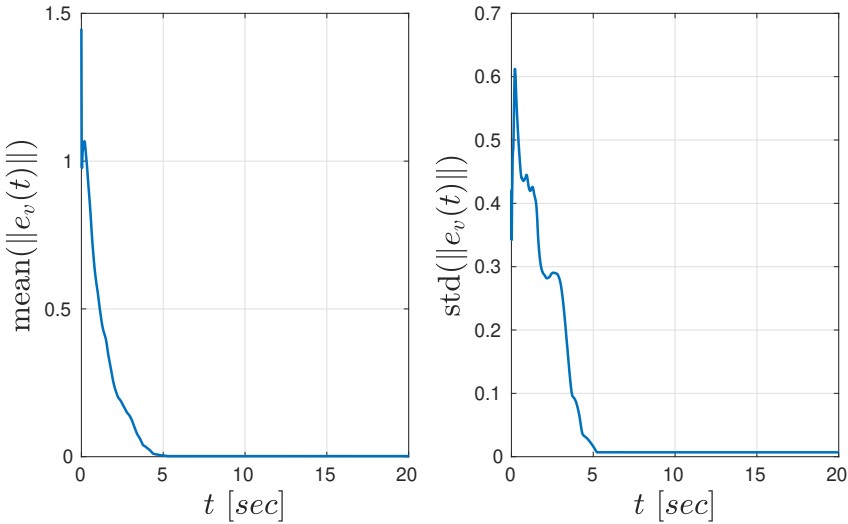

Figure 7: Mean (left) and standard deviation (right) of $e_v(t)$ for the proposed control policy.

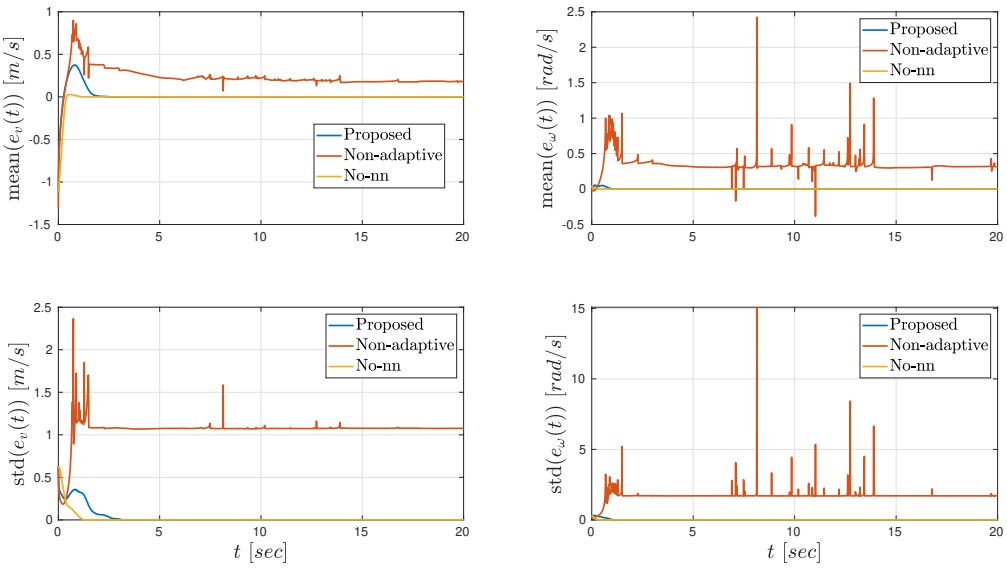

Figure 8: Mean (top) and standard deviation (bottom) of $e_v(t)$ and $e_\omega(t)$ for the proposed, non-adaptive, and no-neural-network control policies.

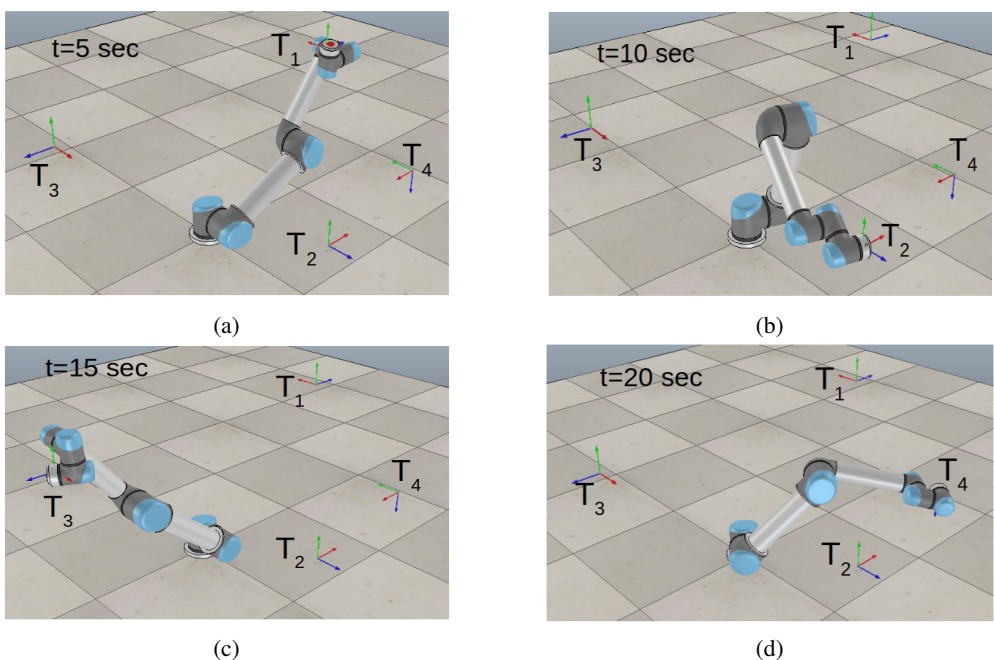

Figure 9: Illustration of the execution of one of the test trajectories of the UR5 robotic arm, visiting the points of interest at the pre-specified time stamps.

Regarding the unicycle experiments, the dynamic terms in (9) have the form

$$M = \begin{bmatrix} M_1 & M_2 \\ M_2 & M_1 \end{bmatrix}$$

$$f_\theta(\bar{x}, t) = d(\bar{x}, t)$$

with $M_1 := \frac{mr^2}{4} + \frac{(I_C + md^2)r^2}{4R^2} + I_0$, $M_2 := \frac{mr^2}{4} - \frac{(I_C + md^2)r^2}{4R^2}$, and where $I_C$ is the moment of inertia of the vehicle with respect to point C (see Fig. 6), $I_0$ is the moment of inertia of the the wheels, and $d$ is the distance of the between point C and the vehicle's center of mass $(p_1, p_2)$. The term $d(\bar{x}, t)$ is chosen as in the robotic-manipulator case. The points to visit are chosen here as $c_1 = [0, 0]^\top$, $c_2 = [0, 2]^\top$, $c_3 = [2, 0]^\top$, $c_4 = [2, 2]^\top$ and $I_i$ is chosen as $I_i = [0, 20]$, $i \in \{1, \dots, 4\}$. We derive 150 problem instances by varying the following attributes: we vary $c_i$ with random offsets in $[-0.3, 0.3]$ and the right end-points of $I_i$ with random offsets in $[-2, 2]$; we add random offsets to the dynamic parameters (elaborated subsequently) and the function $d(\bar{x}, t)$, and we set different sequences of visits to the points $c_i$; we further vary the unicycle's initial position from the first $c_i$ with random offsets in $[-0.3, 0.3]$, and its initial orientation with random offsets in $[-0.25, 0.25]$ (rad) from $\theta(0) = \arctan(e_2(0)/e_1(0))$; we further set random values in $[-0.25, 0.25]$ (rad/s) to the initial wheel velocities.

We set nominal values for the dynamic and geometric parameters, appearing in the inertia matrix, as $m = 28$ (kg), $I_C = 0.1$ (kgm$^2$), $I_0 = 0.01$ (kgm$^2$), $r = 0.01$ (m), $d = 0.01$ (m), and we added random offsets in $(-\frac{m}{2}, \frac{m}{2})$, $(-\frac{I_A}{2}, \frac{I_A}{2})$, $(-\frac{I_0}{2}, \frac{I_0}{2})$, $(-\frac{r}{2}, \frac{r}{2})$, $(-\frac{d}{2}, \frac{d}{2})$, respectively, for the problem instances. We chose the control gains of (12) as $k_d = 0.25$, $k_\beta = 1$, $k_v = k_\omega = k_{\ell_1} = k_{\ell_2} = 10$, $k_v = k_\omega = 1$. The non-adaptive control policy we compared our algorithm with was selected as

$$u_S = M_S \dot{v}_{\text{d}} - k_v e_v + e_d \cos\beta - \beta \frac{\sin\beta}{e_d}$$

$$u_D = M_D \dot{\omega}_{\text{d}} - k_\omega e_\omega + \beta$$

where $v_{\text{d}}$, $\omega_{\text{d}}$ are chosen as (11) and $M_S$, $M_D$ are static estimates of $2\frac{M_1 + M_2}{r}$, $2R\frac{M_1 - M_2}{r}$, respectively. More specifically, we set a deviation of 25% in the parameters appearing in these terms to form $M_S$ and $M_D$. Further experimental results are depicted in Figs. 7-11; Fig. 7b depicts the mean and standard deviation of the velocity errors $e_v(t)$, $e_\beta(t)$ for the 50 test instances, showing convergence to

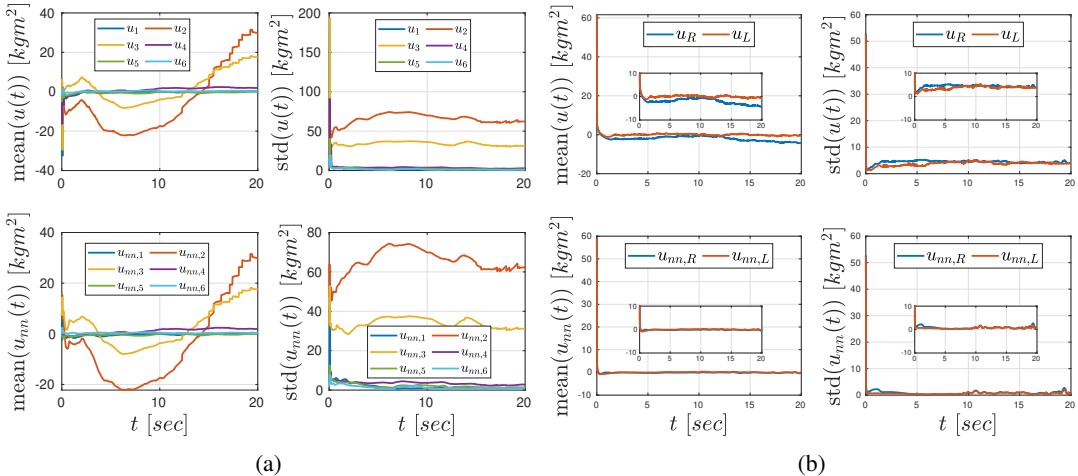

Figure 10: (a): Mean (left) and standard deviation (right) of $u(t)$ (top) and $u_{\mathrm{nn}}(t)$ (bottom) for the proposed control policy and the robotic-manipulator environment. (b): Mean and standard deviation (right) of $u(t)$ (top) and $u_{\mathrm{nn}}(t)$ (bottom) for the proposed control policy and the unicycle environment.

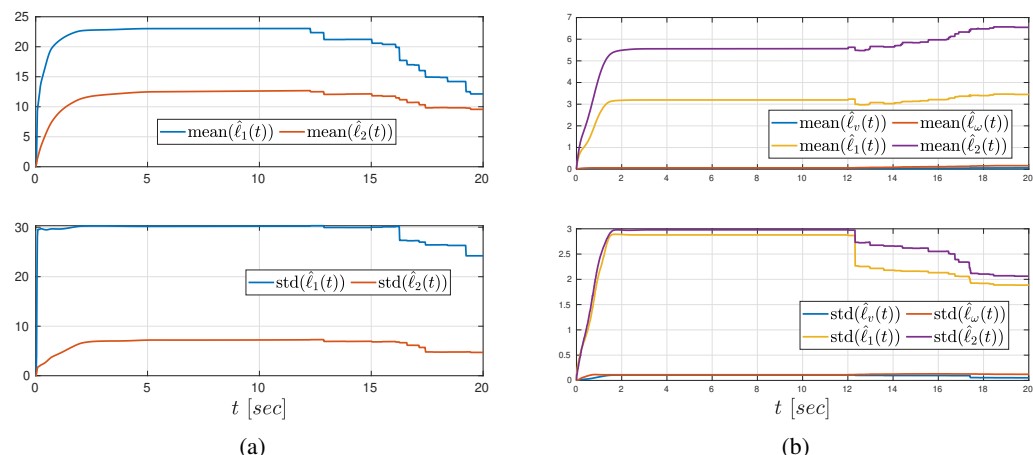

Figure 11: (a): Mean (top) and standard deviation (bottom) of $\hat{\ell}_1(t)$, $\hat{\ell}_2(t)$ for the proposed control policy and the robotic-manipulator environment. (b): Mean (top) and standard deviation (bottom) of $\hat{\ell}_v(t)$, $\hat{\ell}_\omega(t)$, $\hat{\ell}_1(t)$, $\hat{\ell}_2(t)$ for the proposed control policy and the unicycle environment.

zero, whereas Figs. 10b depicts the control input that results from the control policy (4) as well as the the neural-network output. Similarly to the robotic-manipulator case, the control input converges to the neural-network output. Finally, Fig. 11b depicts the mean and standard deviation of the adaptation signals $\hat{\ell}_v(t)$, $\hat{\ell}_\beta(t)$, $\hat{\ell}_1(t)$, $\hat{\ell}_2(t)$ for the 50 test instances.

Finally, regarding the pendulum environment, we consider the dynamics

$$\ddot{q} = \frac{g}{L}\sin q + u + d(t)$$

where $L$ is the pendulum's link length, $g$ is the gravitational constant, and $d(t)$ is a term of exogenous disturbances. We created 150 instances by varying the mass and link length of the pendulum, the external disturbances, and setting its initial position and velocity randomly in $[-1, 1]$ (rad) and $[-1, 1]$ (rad/s), respectively. We selected unitary value for its nominal link length, and we added random offsets in $(-0.5, 0.5)$ in each instance. We set the external disturbances as $d = A\sin(\eta t + \phi)$, with $A$, $\eta$, and $\phi$ taking random values in $[0, 0.2]$, $[0, 1]$, $[0, 2]$, respectively. In contrast to the robotic manipulator case, we assume feedback of $\sin q$, $\cos q$, $\dot{q}$, and set the error $e$ in (3) as $1 - \cos(q - \pi)$. Finally, we chose the control gains as $k_1 = k_2 = 1$, and $k_{\ell_1} = k_{\ell_2} = 10$.

