# OpenReview forum: "Non-Parametric Neuro-Adaptive Control Subject to Task Specifications"
_ICLR.cc/2022/Conference — ICLR 2022 Submitted_

### Official Review · Reviewer_HYqY · 2021-10-19

**Correctness:** 3
**Technical Novelty And Significance:** 2
**Empirical Novelty And Significance:** 2
**Recommendation:** 3
**Confidence:** 4

**Main Review:**

In my opinion, here are the strenght and weakness of the paper.

Strength:
1) by combining control techniques and supervised learning, the learned controller can potentially give a good prior to the online adaptive tracking
2) the presentation of the proposed method is clear.

Weakness:
1) I'm not convinced that the supervised learning is necessary as given the open loop trajectory, there are plenty of existing methods to track it.
2) Theorem 1 does not mean much as the transient tracking can be really bad, thus violating the STL formula.
3) Infinite control input should not be allowed.
4) It seems that if you drop the learning controller, the open-loop trajectory and online adaptive control can still satisfy the system task, making the learning obsolete.

Minor comments:
1) Assumption 1, do you mean the matrix is PSD, or the entries are positive definite? g may not be square
2) Assumption 2 seems trivial as any bounded controller with bounded f and g satisfies that
3) When mixing trajectories under different tasks, there could be states with totally different inputs (for different tasks), does that make the learned controller inferior to even some heuristic controller? Why not include the task information in the learning process?

**Summary Of The Paper:**

This paper presents a learning + adaptive control approach to synthesize control policies that fulfills STL specifications. The open-loop trajectory is generated using (Lindemann & Dimarogonas (2020)), the neural network is trained to replicate the expert demonstration consisting of state-action pairs with trajectories collected following multiple tasks (including the target task). This input is used as the open-loop input and a closed-loop trajectory tracking controller is added to correct the trajectory deviation using adaptive control techniques.



**Summary Of The Review:**

It seems to me that the open-loop trajectory and the online adaptive tracking controller are doing the heavy-lifting here and the learning is not that important. The authors should demonstrate the benefit of learning.

---

> ### Author Response · Authors · 2021-11-22
> **Response to Reviewer HYqY**
>
> We thank the reviewer for their insightful comments. We hereby reply to the reviewer's comments on the paper's weaknesses and minor comments.
>
> Paper weaknesses:
>
> 1. According to the best of our knowledge, there does not exist a methodology for trajectory tracking by systems described by eq. (1) with completely unknown dynamic terms $f$ and $g$. Previous works on adaptive control impose certain assumptions on $f$ and $g$, which we do not. Such assumptions include global boundedness ($|f(x)| <= s$ for a constant $s$), boundedness by known functions ($|f(x)| <= \rho(x)$ for a known  $\rho$) growth conditions ($|f(x)| < k |x|$), and linear parameterizations ($f(x) = W(x) \theta$ - with a known mapping $W(x)$). We do not assume any of the aforementioned conditions. The way we handle the fact that the dynamic terms are unknown is by introducing the neural-network-based learning component, which provides the condition stated in Assumption 2. The intuition behind it is the following: any controller that guarantees the accomplishment of a task will manage to keep the system state $x$ bounded, which, by continuity, implies the boundedness of $f$ and $g$. Therefore, we train the neural network with data from such controllers to approximate a controller that inherits such a property, which motivates Assumption 2.  We have added the aforementioned discussion in the revised version.
>
> 2. Indeed, our main results concern asymptotic stability properties, and therefore the STL formula might not be satisfied. In the revised version, we have excluded the STL tasks from the main focus of the paper, and we now include them only in the experiments.
>
> 3. Note that Theorem 1 guarantees that the control input stays bounded, which is also verified by the experiments. Our future directions include explicitly taking into account specific input-saturation constraints.
>
> 4. As indicated above, without the learning controller, we cannot guarantee the property provided in Assumption 2, and hence the online adaptive control would fail to accomplish the tracking of the open-loop trajectory. Moreover, in the revised version we have conducted new experiments with new system dynamics, which show that online adaptive control performs much better with the learning-controller component than without it.
>
> Minor comments:
>
> 1. By positive definiteness, we imply that $g$ is square with strictly positive eigenvalues. Non-square cases can be also taken into account, but the control design would have to be appropriately modified (as in the case of the unicycle system).
>
> 2. Indeed, if $f$ and $g$ are bounded, Assumption 2 is trivially satisfied. However, we do not assume that $f$ and $g$, or the state $x$, are bounded. In many practical scenarios, $f$ contains quadratic state terms, like, e.g., $x^2$ (e.g., quadratic velocity terms in mechanical systems), which are bounded only if the state is bounded. Intuitively, we expect the neural network to approximate a controller that maintains the boundedness of the dynamics, even when such terms are present. Note that in the experiments, we consider mechanical systems (robotic manipulator and unicycle vehicle) with such unbounded terms in the dynamic model.
>
> 3. The task information could be incorporated into the learning process. Note, however, that we aim to approximate a controller that retains the boundedness of the dynamics, while accomplishing the task at hand via the online feedback-adaptation policy. Therefore, we believe that the learning process can be task-agnostic, focusing on the aforementioned boundedness property. This also results in lower memory requirements, since we do not have to store the task-related information. Finally, a simple heuristic controller might not be able to satisfy Assumption 2 if it does not have any knowledge of the system dynamics (in contrast to the neural-network-based one).

---

> > ### Comment · Reviewer_HYqY · 2021-11-28
> > **Response**
> >
> > I would like to thank the authors for their efforts, but I'm not convinced.
> >
> > 1. IMO, the online adaptive tracking result is more fitting for a control conference as it has nothing to do with learning.
> > 2. I agree, removing STL makes sense.
> > 3. I don't get why theorem 1 guarantees boundedness of inputs, and having bounded input and guaranteeing the closed-loop system satisfying a given input bound are two totally different things. Verified by experiments is not enough for formal claims.
> > 4. Assumption 2 seems merely a Lipschitz condition for the dynamics, which is not difficult to obtain at all, I still don't see the value of learning here.

---

> > > ### Author Response · Authors · 2021-11-29
> > > **Response to Reviewer HYqY**
> > >
> > > Thank you for the comments. We hereby reply to each one of them separately:
> > >
> > > 1. We agree with the reviewer, the adaptive tracking result fits better at a control conference. However, we believe that its combination with the learning component makes it suitable for the ICLR conference.
> > >
> > > 2. Thank you for the comment.
> > >
> > > 3. Note that, in the proof of Theorem 1, the function $V$ is non-increasing (since its derivative is proven to be non-positive). That means that $V(t)$ is bounded by its initial value, i.e., $V(t) \leq V(0)$ (with a slight abuse of notation, we express it as a function of time). By inspecting $V$, the latter implies that the terms $e_v$, $\hat{\ell_1}$, $\hat{\ell_2}$, appearing in the control policy (4) remain also bounded. Additionally, by definition, the neural-network controller $u_{nn}$ is also bounded. Therefore, all the terms of (4) remain bounded, proving the boundedness of the control input. As the reviewer correctly notices, however, bounded input and satisfaction of a given bound are different things. Currently, the proposed algorithm does not guarantee the satisfaction of an explicit input bound. This can be potentially achieved via tuning of the control gains $k_1$, $k_2$, $k_{\ell_1}$, and $k_{\ell_2}$. In particular, smaller values of these gains lead to smaller control input, but also smaller rates of convergence of $e$ and $e_v$.  An explicit relation among an upper bound for the control input and the control gain $k_2$ (of the form $\|u\| \leq \bar{u}_1 + k_2\bar{u}_2$) can be obtained by using the inequality $V(t) \leq V(0)$, obtained in Theorem 1. The control gain $k_2$ can then be tuned in order to achieve a desired bound $u_d$. Such tuning, however, requires knowledge of the bounds of the dynamic terms in equation (2).
> > >
> > > 4. Thanks for the comment. Consider a second-order dynamical system of the form of eq. (1) with $f(x) = -x_2^2 + m$, and $g(x) = 1$, where $m$ is some gravitational constant.  Such a function $f(x)$ is very common in mechanical systems, consisting of Coriolis effects and gravity terms. Note that in that case, the condition in Assumption 2 can only be satisfied if the function $f(x)$ is somehow compensated for by the controller $u_{nn}$, which would require explicit knowledge of $f(x)$ (for instance $u_{nn} = -f(x) - x$). In any other case, the quadratic term $x_2^2$ cannot be bounded by the linear terms $\|x\|$ or the constant of the right-hand-side of eq. (2).
> > > However, we do not assume the availability of a continuous controller $u_{nn}$ that is aware of $f(x)$ and hence can accurately compensate for it. We are motivated by cases where systems undergo changes in their dynamics and hence we assume the availability of discrete data from controllers that compensate for different variations of the dynamics (and hence of $f(x)$), and hence are able to guarantee the boundedness of the respective closed-loop systems. The neural-network learning is therefore needed in order to (1) output a continuous function from this discrete data, and (2) approximate a controller that learns this property of maintaining the boundedness of the dynamics as per Assumption 2 (eq. (2)).

---

### Official Review · Reviewer_GDik · 2021-11-02

**Correctness:** 4
**Technical Novelty And Significance:** 1
**Empirical Novelty And Significance:** 1
**Recommendation:** 5
**Confidence:** 4

**Main Review:**

*** STRENGTHS ***

(1) The paper is generally well-written, and is timely given a recent surge in works at the intersection of parametric/non-parametric learning and adaptive control theory.

(2) The experiments in the paper are extensive; it is good to see multiple different robotic systems included. The experimental results seem detailed and reproducible.

*** WEAKNESSES ***

The core algorithmic content of the paper is spread across: (1) learning a nominal feedback law as a neural network to bound the system dynamics, (2) computing open-loop trajectories satisfying SITL formulae, and (3) adaptive controller design.

Regarding (1): It seems naive to train a single feedback law on data from many different tasks and models (e.g., different controllers are often used for different system dynamics). Moreover, the authors provide no convincing evidence (rigorous, intuitive, or otherwise) that their learned neural network feedback law should satisfy their Assumption 2. Even if the neural network is trained on bounded trajectories, this does not guarantee that the closed-loop dynamics satisfy the particular affine bound in Assumption 2. Overall, the neural network learning content of the paper is not substantive and not convincingly conducive to the adaptive control theory presented later on.

Regarding (2): the paper's narrative is muddied by its focus on SITL. In section 3.2, the authors describe that, using _existing_ work, a smooth open-loop trajectory in R^n satisfying all of the requirements of a SITL task can be computed. Given that the authors do not make any contributions to SITL work and just reduce any SITL task to a trajectory tracking problem, sections 2.1 and 3.2 should be removed entirely. If desired, the authors could simply include a comment on applications of adaptive trajectory tracking control to SITL tasks in a motivation or discussion section.

So far, the possible algorithmic contribution of the paper has been narrowed down to the design of the adaptive controller on pages 5-6 (assuming any nominal controller that bounds the dynamics according to Assumption 2). However, the adaptive controller design contribution is not adequately contextualized against the adaptive control literature. Other than citing an adaptive control textbook from Krstic, et al, the authors need to cite related works and establish the novelty of their design. For example, Equations (3)-(4) seem reminiscent of well-known sliding-mode adaptive controllers, e.g., from:

    J.-J. E. Slotine and J. A. Coetsee. "Adaptive sliding controller synthesis for non-linear systems". International Journal of Control, 1986.

    J.-J. E. Slotine and W. Li. "On the adaptive control of robot manipulators". International Journal of Robotics Research, 1987.

Indeed, the authors' "e_v" is exactly Slotine, et al's sliding variable "s", and the authors' "v_d" is Slotine, et al's reference velocity "q_r". The authors need to discuss how their design differs from such well-known existing work.

Finally, the content on the adaptive control design for a unicycle system, which spans over a page, seems ill-fitted to the main content of an ICLR paper submission.

**Summary Of The Paper:**

In this paper, the authors propose an adaptive controller that can be applied to a certain class of dynamical systems in order to fulfill Signal Interval Temporal Logic (SITL) tasks. The proposed controller leverages a nominal feedback law learned from past trajectory data collected on possibly many different SITL tasks and system parameters. The authors test their method in simulation on a manipulator arm, unicycle robot, and a pendulum with varying physical parameters and external forces.

**Summary Of The Review:**

Of the three sections in the "Main Results" of the paper, only the adaptive control design section is substantive; the proposed neural network training is just naive regression on state-input tuples, and the inclusion of SITL needlessly complicates the paper's exposition before being reduced to a trajectory tracking task. Adaptive control theory alone may be an unsuitable fit for ICLR. Regardless, the novelty of the adaptive control design is questionable, as it appears similar to well-known previous work and the authors' have not contextualized their design amongst the adaptive control literature.

%%
Note: I have raised my score after the authors' rebuttal.

---

> ### Author Response · Authors · 2021-11-22
> **Response to Reviewer GDik**
>
> We thank the reviewer for their insightful comments. We hereby reply to the reviewer's comments on the paper's weaknesses.
>
> Regarding (1):
> We are inspired by cases where systems undergo modifications (e.g., the substitution of a motor or link in a robotic arm, exposure to new working environments, or gradual faults) that change their dynamics in a way that prevents them from using the already endowed controllers that accomplish certain tasks. In such cases, instead of identifying the new dynamics and re-designing new controllers, we exploit the existing ones (via the neural-network learning component), which might correspond to different dynamic models and tasks, and use the proper online adaptation policy to handle the residual error. Note that, in the training data, the common feature of all the training controllers is that they keep the dynamics bounded since they achieve tasks that correspond to bounded trajectories. The neural network aims to approximate a controller that retains this property, which leads to Assumption 2. Such as assumption is motivated by the property of neural networks to approximate continuous functions. More specifically, the universal approximation property states that a neural network can approximate a continuous function arbitrarily well in a compact set. In fact, such property has been used several times in many related works. In our case, we require something more relaxed than an arbitrarily good approximation of a certain function. Instead, we assume that the neural network is able to approximate a controller sufficiently well to keep the dynamics bounded by the linear term in the right-hand side of the inequality of Assumption 2 - which is not an as strong assumption as an accurate approximation of a specific function.  In the experiments, we observed that the assumption was satisfied along the testing trajectories of the closed-loop system. Future work will focus on further elaborating on whether or which parts of the state space it holds.
>
> Regarding (2):
> Indeed, in the revised version, we have changed the focus of the paper towards general tasks encoded by reference trajectories, and removed the SITL parts; we only consider SITL tasks in the experiments.
>
> Moreover, following an adaptive-like methodology, our control design is indeed similar to many related works in the literature. However, all these works impose certain conditions on the system dynamics, which we do not. For instance, the work [Slotine et al, “On the adaptive control of robot manipulators”, IJC, 1986] assumes that either the dynamics are linear, or the dynamic terms of the system are bounded by known functions, which are then explicitly used in the adaptive control design; the work [Slotine et al, "On the adaptive control of robot manipulators", IJRR, 1987] assumes linear parameterization of the dynamic terms, whose structure is used in the control design. Other commonly found assumptions in the related literature consist of globally bounded dynamic terms or linear growth conditions (see the newly added references in the revised version of the paper). In many practical situations, however, systems do not satisfy such conditions. A standard example is the term $x^2$, which appears frequently in the dynamics of mechanical systems (quadratic velocity term), and does not satisfy global boundedness or linear growth conditions. In addition, as mentioned above, systems might undergo certain failures that modify the underlying dynamics and render thus any a priori information on the dynamics not usable. Note that we do not impose such structural assumptions on the dynamics terms and we do not employ any approximation schemes. Assumption 2 is essentially a condition on the neural network, motivated by the universal approximation property of such structures.
>
> In the revised version, we have added comments related to the aforementioned discussion. Finally, we have placed the control design for the unicycle system in the Appendix.

---

> ### Comment · Reviewer_GDik · 2021-11-30
> **Feedback**
>
> I would like to thank the authors for their thoughtful responses to my comments. I think the revised version of the paper is clearer and thus I have raised my score for this paper. However, I share YVCm reviewer's concern that the "corrections weaken the claims."

---

### Official Review · Reviewer_YVCm · 2021-11-04

**Correctness:** 3
**Technical Novelty And Significance:** 3
**Empirical Novelty And Significance:** 3
**Recommendation:** 5
**Confidence:** 3

**Main Review:**

Strengths
1. The approach presented in this paper is novel and differs significantly from past proposals.
2. The approach is demonstrated on robotic applications.
3. The concept of using a learned component within a control task is of interest to this community.

Weaknesses
1. The paper claims that the proposed method can be used to achieve behaviors of the controlled system which satisfy SITL specifications. This appears to be subtely incorrect. Theorem 1 describes asymptotic performance guarantees, not performance of the system during a finite horizon. Nothing can be said about the satisfaction of the "prefix" portion of the SITL specification.
2. The paper correctly claims that the theorem makes no special requirements about the error of the learned component, and moreover that the selection of training data does not affect the claim. This is not the same thing as saying that these elements "don't matter", what is promised is that things won't go terribly wrong, not that the outcomes will be good. The experimental section could be improved by ablations of the off-policy data both in terms of variations of the system dynamics, the quantity of the data, and the diversity of tasks.
3. The learning problem in which the learned component predicts the next control input seems like it could have issues (which do not affect the theoretical guarantees) in the case that there are two tasks with the same "prefix" trajectory but a different end trajectory. It seems that the network would then be forced to predict the average of the two outcomes. Does the methodology seek to avoid sampling such training data. What are the implications?
4. In general the paper oversells some of the claims. See above. it could be improved by taking a less adversarial approach in the writing and acknowledging the trade off in relaxing the approach to ask that the learned component doesn't add arbitrarily large errors. In addition more discussion could be added about the controllability requirement and how limiting this is; I acknowledge that the paper has made some efforts to be precise here.
5. A sketch of the proof in the main body could be useful. It is extremely technical. Moreover, it could be useful for the author's to discuss whether the learned component must be a neural network and specific properties of neural networks were necessary for the proof. It seems that the setup could be more general.
6. The use of SITL seems needlessly complex and not the main point of the paper. Beyond its use as a specification language for generating training examples it seems ancillary. See point 1. The paper could be improved by incorporating specifications that use more properties of the SITL formalism for task specification (e.g. beyond reach like tasks).
7. In the related work some discussion of work like: https://arxiv.org/pdf/1903.11239.pdf could be included. In this reference the authors also learn a component to augment control inputs. They do not have any similar guarantees but it is a better motivated setting.

**Summary Of The Paper:**

The paper describes an adaptive controller which includes a learned component. The paper proves that this controller is asymptotically stable. Importantly the paper develops theory which proves that it is not necessary for the learned component to achieve bounded error on the task of predicting the control input given the state, but instead only that the output of the combined system be bounded by a simple function of the current state. The method incorporates a proposal for generating training data for the learned component by sampling trajectories (open-loop) of the system which conform to an SITL specification. The method is evaluated on several common simulation environments which comprise "reach-like" motion planning tasks. The results demonstrate that the proposes adaptive controller converges to the desired reference trajectory, sometimes faster than adaptive methods which do not incorporate the learned component.

**Summary Of The Review:**

This paper describes an approach to adaptive control incorporating a learned component. The novelty relies in the development of a proof technique which doesn't require the learned component to have Lipschitz like properties or exhibit universally bounded error on a space of inputs. The paper is interesting to the community and combines theory with experiment in a coherent manner.

---

> ### Author Response · Authors · 2021-11-22
> **Response to Reviewer YVCm**
>
> We thank the reviewer for their insightful comments. We hereby reply to the reviewer's comments on the paper's weaknesses.
> 1. We agree with the reviewer. We have appropriately modified the revised version to mainly focus on the trajectory-tracking problem, and we use SITL tasks only in the experiments. Note, however, that in practice many tasks are encoded as trajectory-tracking problems, and the transient behavior of the system is handled by appropriate tuning of the control gains.
> 2. Indeed, the learned component is only required to maintain the boundedness of the dynamics by the growth condition in Assumption 2. We are inspired by cases where systems undergo modifications (e.g., the substitution of a motor or link in a robotic arm, exposure to new working environments, or gradual faults) that change their dynamics in a way that prevents them from using the already endowed controllers that accomplish certain tasks. In such cases, instead of identifying the new dynamics and re-designing new controllers, we exploit the existing ones (via the learning component) and use the proper online adaptation policy to handle the residual error. The existing controllers might often come from systems with variations in the dynamic model. Hence, in the experimental section, we use training data that correspond to a variety of different tasks, models, and controllers. Additionally, the controllers that correspond to these tasks and models have the property of maintaining the boundedness of the system dynamics; the learned component aims to approximate a controller with such a property, leading to Assumption 2.
> 3. Indeed the learning algorithm aims to approximate an average controller, in terms of the executed tasks. Note that the controllers of the training data guarantee the satisfaction of the respective tasks, which implies that these controllers manage to guarantee the boundedness of the system dynamics. The learning component aims to approximate a controller that maintains this property, which leads to the condition of Assumption 2. One could also train the system only in the desired test task to be achieved, which however would be overly conservative; we use multiple training tasks to show the ability of the algorithm to “generalize”, i.e.,  perform well in tasks that were not used in the training. Additionally, as explained in our response to the previous comment, we are motivated by cases where model changes prevent the system from using its existing controllers. In such cases, we wish to exploit all the available offline data, which might correspond to several different tasks and models. Moreover, the neural-network controller is only required to maintain the boundedness of the system dynamics (as per Assumption 2), which is achieved by all the training controllers. We have emphasized that in the revised version.
> 4. In the revised version of the paper, we have added a discussion on the addition of large errors by the learned component; in practice, this property depends on the performance of the underlying learning algorithm - in this case, the learning capabilities of the underlying neural network. The controllability requirement is satisfied by many engineering systems and its introduction is a way for providing a general control algorithm for this class of systems. Systems that do not satisfy the controllability requirement, such as nonholonomic or underactuated systems,  need a more specific control design that exploits the system’s structure. An illustrating example is the control algorithm we design for the unicycle vehicle.
> 5. In the revised version we have added a sketch of proof, and we have moved the unicycle control in the appendix (requested by reviewer 2). Indeed, as the reviewer correctly notices, the learned component does not necessarily need to be a neural network. However, the use of neural networks is motivated by Assumption 2, which is related to the approximation of the training controllers.
> 6. In the revised version we use SITL only in the experiments. Richer SITL specifications are part of our future work.
> 7. Thanks for the suggestion, we have included it in the revised version.

---

> > ### Comment · Reviewer_YVCm · 2021-11-29
> > **Comments on the updated draft**
> >
> > Thank you for the updated draft and clear response to my comments. I enjoyed reading this paper, and the discussion with the authors.
> >
> > 1. The new title, abstract, edits, and proof sketch are improvements.
> >
> > 2. Regarding point 2 above, I would like to reiterate that I think re-running the experiments with different amounts of training instances and different random samples of the each robot configuration would be useful. Since the proof says that "terrible things won't happen" not that "good things will happen" the experiments should work harder to demonstrate empirically how training data influences performance in a positive way.
> >
> > 3. More literature in the vein of "residual physics/control" or PILCO style work would be useful given that this is a learning focused conference.
> >
> > 3. Although the I'm satisfied with the responses to my questions, the corrections weaken the claims. As such I'm choosing to leave my score unchanged.

---

### Author Response · Authors · 2021-11-22
**Change of title and abstract**

We have modified the title and abstract of the revised version of the paper to:

Title:  NON-PARAMETRIC NEURO-ADAPTIVE CONTROL

Abstract:

We develop a learning-based algorithm for the control of autonomous systems governed by unknown, nonlinear dynamics to satisfy user-specified tasks expressed via time-varying reference trajectories. Most existing algorithms either assume certain parametric forms for the unknown dynamic terms or resort to unnecessarily large control inputs in order to provide theoretical guarantees. The proposed algorithm addresses these drawbacks by integrating neural-network-based learning with adaptive control. More specifically, the algorithm learns a controller, represented as a neural network, using training data that correspond to a collection of system parameters and tasks. These parameters and tasks are derived by varying the nominal parameters and the reference trajectories, respectively. It then incorporates this neural network into an online closed-form adaptive control policy in such a way that the resulting behavior satisfies the user-defined task. The proposed algorithm does not use any a priori information on the unknown dynamic terms or any approximation schemes. We provide formal theoretical guarantees on the satisfaction of the task. Numerical experiments on a robotic manipulator and a unicycle robot demonstrate that the proposed algorithm guarantees the satisfaction of 50 user-defined tasks, and outperforms control policies that do not employ online adaptation or the neural-network controller. Finally, we show that the proposed algorithm achieves greater performance than standard reinforcement-learning algorithms in the pendulum benchmarking environment.

---

### Decision · Program_Chairs · 2022-01-20

**Decision:**

Reject

**Comment:**

The reviewers acknowledge that the paper is well written and contains interesting ideas to combine adaptive control and learning. However, they identified issues regarding the claims about transient tracking and STL formula. Moreover, the significance of the presented learning rule was unclear regarding one reviewer. While the authors could respond well to the identified transient tracking issue, they also needed to weaken their claims, limiting the contribution of the paper. The reviewers therefore stayed with a a reject rating.